# Shifts in ophthalmic care utilization during the COVID-19 pandemic in the US

Charles Li [1✉], Flora Lum[1], Evan M. Chen[2], Philip A. Collender [3], Jennifer R. Head [3], Rahul N. Khurana[2,4], Emmett T. Cunningham Jr.[5,6,7], Ramana S. Moorthy[8,9], David W. Parke II [1] & Stephen D. McLeod[1,2,6]

## Abstract

**Background** Healthcare restrictions during the COVID-19 pandemic, particularly in ophthalmology, led to a differential underutilization of care. An analytic approach is needed to characterize pandemic health services usage across many conditions.

**Methods** A common analytical framework identified pandemic care utilization patterns across 261 ophthalmic diagnoses. Using a United States eye care registry, predictions of utilization expected without the pandemic were established for each diagnosis via models trained on pre-pandemic data. Pandemic effects on utilization were estimated by calculating deviations between observed and expected patient volumes from January 2020 to December 2021, with two sub-periods of focus: the hiatus (March-May 2020) and post-hiatus (June 2020–December 2021). Deviation patterns were analyzed using cluster analyses, data visualizations, and hypothesis testing.

**Results** Records from 44.62 million patients and 2455 practices show lasting reductions in ophthalmic care utilization, including visits for leading causes of visual impairment (age-related macular degeneration, diabetic retinopathy, cataract, glaucoma). Mean deviations among all diagnoses are 67% below expectation during the hiatus peak, and 13% post-hiatus. Less severe conditions experience greater utilization reductions, with heterogeneities across diagnosis categories and pandemic phases. Intense post-hiatus reductions occur among non-vision-threatening conditions or asymptomatic precursors of vision-threatening diseases. Many conditions with above-average post-hiatus utilization pose a risk for irreversible morbidity, such as emergent pediatric, retinal, or uveitic diseases.

**Conclusions** We derive high-resolution insights on pandemic care utilization in the US from high-dimensional data using an analytical framework that can be applied to study healthcare disruptions in other settings and inform efforts to pinpoint unmet clinical needs.

## Plain language summary

The COVID-19 pandemic disrupted healthcare services globally, including eye care in the United States. Using a US eye disease database, we measured how the pandemic impacted patient visits for 261 eye diagnoses by comparing actual visit volumes for each diagnosis with what would have been expected without the pandemic. We identified groups of conditions with similar changes in visit levels and examined whether these shifts were related to characteristics of the diagnoses studied. We found extended decreases in patient presentations for most eye conditions, with greater reductions for less severe diagnoses, and with anomalies and differences in this trend across diagnosis categories and pandemic sub-periods. This highlights areas of potentially unmet need in vision care arising from the pandemic.

[1] American Academy of Ophthalmology, San Francisco, CA, USA. [2] Department of Ophthalmology, University of California, San Francisco, CA, USA. [3] Division of Environmental Health Sciences, School of Public Health, University of California, Berkeley, CA, USA. [4] Northern California Retina Vitreous Associates, Mountain View, CA, USA. [5] Department of Ophthalmology, California Pacific Medical Center, San Francisco, CA, USA. [6] The Francis I. Proctor Foundation, UCSF School of Medicine, San Francisco, CA, USA. [7] Department of Ophthalmology, Stanford University School of Medicine, Stanford, CA, USA. [8] Associated Vitreoretinal and Uveitis Consultants, Carmel, IN, USA. [9] Department of Ophthalmology, Indiana University Medical Center, Indianapolis, IN, USA. ✉email: cli@aao.org

During the COVID-19 pandemic, sustained disruptions to essential health services were documented worldwide[1,2]. In the United States, historic declines in care utilization were widely observed during the pandemic's acute phase in Spring 2020. Despite subsequent recoveries in patient volume and healthcare spending, many measures of overall healthcare utilization failed to return to[3,4], or exceed[2,5], pre-pandemic levels. Among all specialties, ophthalmology has experienced one of the most severe disruptions to care[4,6,7], likely related to the specialty's reliance on close-proximity examinations, an older patient population, and a large proportion of elective procedures[8,9]. Studying the patterns of ophthalmic care utilization during the pandemic may therefore not only reveal emerging areas of potentially unmet need in population vision health, but also contribute further insights on key characteristics associated with services that patients and health systems may regard as priorities. We therefore conducted a high-dimensional study to understand shifts in eye care utilization patterns for a comprehensive set of ophthalmic diagnoses during the first two years of the COVID-19 pandemic in the United States.

Redistributions of essential healthcare resources during the acute phase of the pandemic were required to minimize mortality, but the patterns of, and reasons explaining, sustained utilization reductions in the post-acute phase are not entirely clear[10]. Previous studies by health economists have formally estimated the responsiveness, or elasticity, of demand of healthcare services to changes in cost or income[11–13]. For instance, emergency room visits tended to exhibit little change in demand in response to changes in price, whereas pharmaceuticals, mental health/substance abuse treatment, and specialist care had high elasticities of demand[11]. Similarly, we explored how utilization levels for a wide range of ocular diagnoses exhibited varying degrees of sensitivity to possible pandemic-related restrictions to the seeking or delivery of care (e.g., resource constraints, behavioral modifications). We specifically investigated possible factors driving the differential underutilization of ophthalmic care during the pandemic by examining whether characteristics of medical problems themselves—namely, disease severity—were associated with observed changes in care utilization relative to levels expected in the absence of the pandemic.

Therefore, unlike previous studies[4,14–16] that investigated how pandemic utilization trends differed by patient demographic characteristics or the setting and type of care, or ones that studied overall categories of diagnoses or a limited selection of medical problems[17], our investigation explored the variations in healthcare usage across a comprehensive spectrum of detailed diagnoses from a single specialty via a common analytical framework that related care utilization patterns to specific attributes of medical problems. Using electronic health record (EHR) data from a large national disease registry inclusive of most ambulatory ophthalmic care in the United States, we extracted time series data for each ocular condition included in the study, generated predictive models from pre-pandemic data to establish counterfactual levels of healthcare utilization for each condition expected in the absence of the pandemic, estimated pandemic effects on utilization by calculating deviations of observed data from predictions, and investigated characteristics of conditions with similar deviations. In doing so, we illustrate an analytic approach that can be adapted to different settings to serve as an initial tool for an exploratory, but expansive and granular, assessment of impacts to care utilization during prolonged disruptions to healthcare access. The high-resolution insights (i.e., derived from detailed ophthalmic diagnoses) generated from this high-dimensional analysis (i.e., conducted across an expansive range of ocular conditions, and over different subperiods and all months spanning the first two years of the pandemic) may inform future studies aiming to determine the clinical impacts of missed or delayed care in specific patient populations and disease cohorts, monitor the pandemic's effect on healthcare access, and clarify distinctions between harmful and benign reductions to care. Furthermore, because this study is, to our knowledge, the largest and most comprehensive characterization of ophthalmic care patterns during COVID-19, its findings may also contribute to a developing understanding of the pandemic's long-term impact on population vision health.

In our analysis of ophthalmic visits from over 44 million patients seen across 2455 practices in the United States, we find significant and enduring reductions in patient presentations across most of the 261 diagnoses that are included for examination in this study. These consistent declines in visits are generally more pronounced for less severe ocular conditions, including diagnoses that represent the least critical forms of the leading causes of visual impairment and blindness in the US. We also identify clusters of diagnoses with exceptional deviations in visit patterns over time, highlighting conditions exhibiting above- and below-average changes in care utilization during the pandemic. Intense reductions in visits mostly occur for non-vision-threatening conditions like conjunctivitis, or precursors of vision-threatening conditions like subclinical indications of diabetic retinopathy. In contrast, conditions with above-average visits include emergencies posing a risk for irreversible vision loss, such as retinopathy of prematurity (ROP). These findings underscore the widespread effect of the pandemic on eye care services in the US, emphasizing the need for more targeted insights of its long-term clinical impacts.

## Methods
Patterns in care utilization were characterized using a common analytical framework, which consisted of multiple stages (Fig. 1).

**Inventory of diagnoses**. We constructed an expansive inventory of ocular conditions to study by grouping more than 3300 ophthalmic ICD-10 diagnosis codes into 336 clinically meaningful diagnosis entities adapted from categorizations provided by the US Agency for Healthcare Research and Quality Clinical Classifications Software Refined (CCSR) database version 2020.2[18]. We followed a common set of considerations to modify these categorizations of ICD-10 codes where needed for the analytic purposes of this study, and to further create more granular groupings of these ICD-10 codes into diagnosis entities (Supplementary Note 4). For instance, all diagnosis entities were required to have a sufficient utilization level (i.e., above single-digit patient counts) for each month of the study period (January 2017–December 2021). After establishing an initial set of 336 diagnosis entities, to ensure that study findings were based on reliable predictions of utilization levels expected in the absence of the pandemic, conditions with poor counterfactual model performance (considered as ≥12.5% root-mean-squared percentage error [RMSPE]) were also excluded, resulting in a final set of 261 diagnosis entities attributed to 13 mutually exclusive diagnosis categories (e.g., blindness and vision defects, cataract and other lens disorders, corneal and external disease) (Supplementary Data 1) included for analysis in this study.

**Data extraction and study setting**. Monthly numbers of patients documented with each diagnosis entity were queried from the American Academy of Ophthalmology IRIS® Registry (Intelligent Research in Sight), the United States' first comprehensive clinical database of eye diseases, encompassing over 440 million patient visits from 73 million unique patients and 3000 practices as of April 1, 2022. The IRIS Registry consists of deidentified data from EHRs submitted on a daily or weekly basis by practices representing over 70% of US ophthalmologists, and contains records on ocular diagnoses, procedures, visits, examinations, medications; patient demographic variables; and the setting of care.

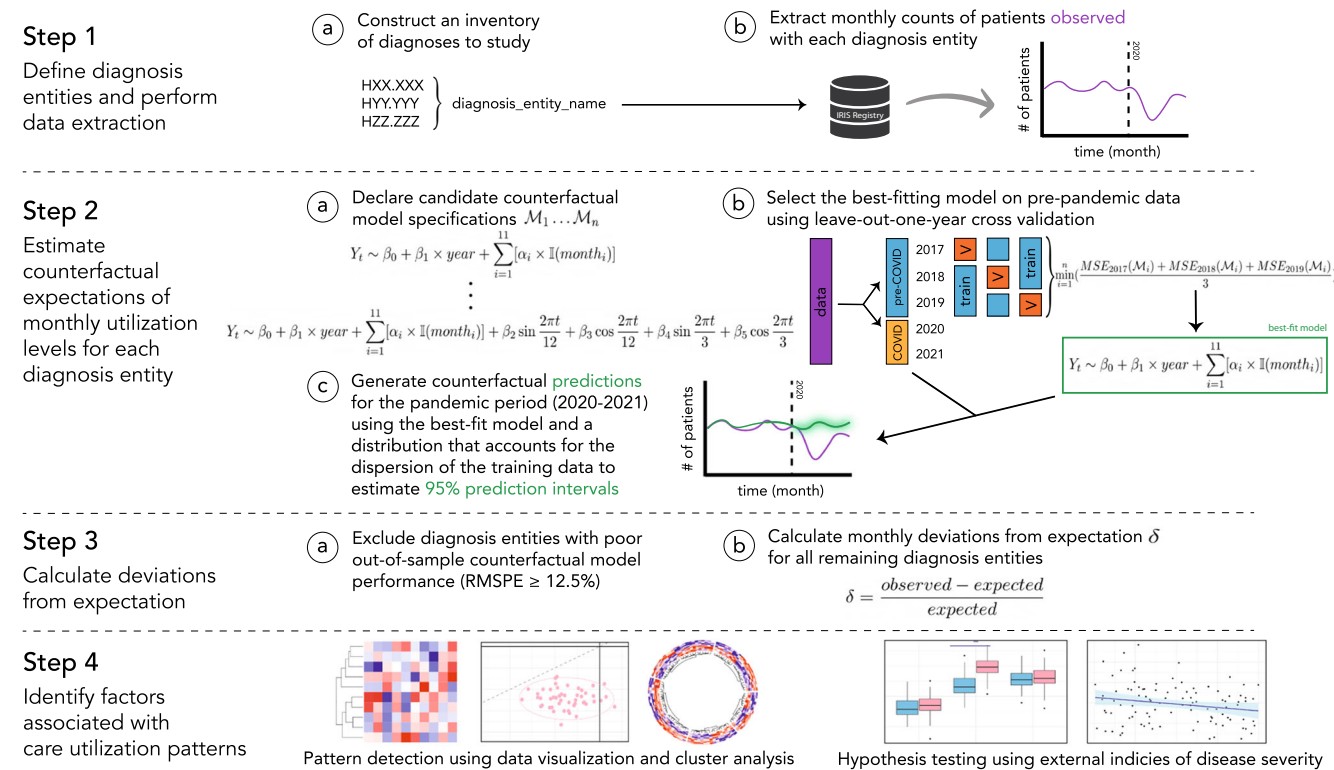

**Fig. 1 A common analytical framework for the high-dimensional characterization of care utilization patterns across many conditions.** Full specifications of candidate counterfactual models (Step 2a) are available in Supplementary Note 1. *Abbreviations*: IRIS Registry American Academy of Ophthalmology Intelligent Research in Sight Registry®, RMSPE root-mean-squared percentage error, MSE mean squared error.

Although its coverage is not as large as with private, outpatient practices, the IRIS Registry, which is compatible with multiple EHR software vendors[19], is also integrated with approximately 33% of Association of University Professors of Ophthalmology member academic institutions. The methods and operations of the IRIS Registry have been previously described[20]. This study did not require ethical approval or informed consent because the IRIS Registry contains deidentified patient data.

We defined a global study period of January 1, 2017, to December 31, 2021, and delineated a pre-pandemic period (2017–2019) that was used to establish counterfactual baselines for patient volumes, and a pandemic period (2020–2021) for which deviations from expectation were calculated. Within the pandemic period, we defined a hiatus period (March–May 2020) to highlight the steepest reductions in utilization during the acute phase of the pandemic, and a post-hiatus period (June 2020–December 2021) to represent the post-Spring 2020 recoveries that were observed widely in our analysis and elsewhere[3–5]. Patients included in the monthly count for a diagnosis entity were documented with a physician encounter (a procedure, ocular examination, or visit record) on the same date as the diagnosis. Furthermore, to exclude from consideration changes in utilization trends that may be attributable to new openings, permanent closures, or other changes in the data reporting statuses of practices, we only analyzed EHRs from practices that reported data to the IRIS Registry throughout all months of the global study period.

**Establishing counterfactual levels of pandemic care utilization.** A parametric approach was used to estimate monthly numbers of patients expected for each diagnosis entity in the absence of the pandemic. We fit predictive models to time series data from the pre-pandemic period and used the fitted models to predict counterfactual utilization levels during the pandemic. In particular, for each condition, we used a leave-out-one-year blocked cross validation algorithm to select a best-fitting model, among a pre-specified set of candidate models, that had the best predictive ability, which was determined by identifying the candidate model with the lowest average mean squared error across all holdout years from the pre-pandemic period[21,22]. Candidate predictive generalized linear models initially assumed conditionally Poisson-distributed outcomes and included combinations of terms for seasonality (monthly fixed effects and/or harmonic terms with 3-, 6-, and 12-month periodicities) and a linear trend over years (Supplementary Note 1). Finally, to establish counterfactual levels of care utilization during the pandemic period, errors in the fit of the model selected for each diagnosis entity were assessed for overdispersion, after which 100,000 Monte Carlo-simulated predictions were generated from conditional Poisson, over-dispersed Poisson, or negative binomial distributions in accordance with the assessed mean-variance relationship of model predictions to residual values. 95% prediction intervals (PIs) for monthly counterfactual predictions were subsequently calculated (Supplementary Note 2). For each diagnosis entity, the specification of the selected model, along with model performance metrics and descriptive summary statistics, are provided in Supplementary Data 2.

**Measures of pandemic effects on care utilization and time to recovery.** Our primary outcome, $\delta$, was, for each diagnosis entity, its deviation from expected care utilization, calculated as the relative difference between observed and expected numbers of patients for a given month:

$$\delta = \frac{observed - expected}{expected} \qquad (1)$$

For each deviation, 95% confidence intervals (CIs) and two-sided empirical p-values were obtained by referencing the

counterfactual distribution of the monthly prediction. We also estimated average deviations across summary timeframes: for each quarter (Q3 2020 to Q4 2021), and across all months, of the post-hiatus period. Average deviations were computed by taking the mean of the joint distribution of deviations across each month in the summary period, and 95% CIs and two-sided empirical p-values were similarly obtained.

As a secondary outcome, we examined the time it took for utilization to recover to expected values. Recovery was defined as three or more consecutive months for which no statistically significant negative deviations from expectation ($\delta < 0$, adjusted $p \leq 0.05$) were recorded. Among conditions that recovered, sustained recovery described those that did not experience further significant negative deviations, and partial recovery for the conditions that did.

**Analysis of care utilization patterns and their association with diagnosis severity.** To identify groups of diagnosis entities with common patterns in monthly deviations over time, we applied a hierarchical clustering algorithm (using the Euclidean distance function and the complete linkage method[23]) and generated cluster heatmaps[24–26]. To examine associations between condition severities and utilization patterns, we relied on both qualitative insights derived from visualizations of condition deviations throughout the pandemic, and hypothesis tests that used disease severity measures provided by prior studies. Based on a global survey of over 70 experts on the severities of common ocular emergencies, Bourges et al. aggregated severity rankings for each surveyed condition, ranging from 1 (least severe) to 5 (most severe)[27]. We mapped 36 diagnosis entities from our study to the set in Bourges et al. (Supplementary Table 1), and examined associations between severity and utilization at different time points using univariable linear regressions. Similarly, using Wilcoxon rank-sum tests, we investigated whether significant differences in deviations existed between vision-threatening (VT) and non-vision-threatening (NVT) subtypes of the leading causes of visual impairment in the US[28]: diabetic retinopathy (DR), age-related macular degeneration (AMD), and glaucoma (Supplementary Table 2).

$P$-values $\leq 0.05$ were considered statistically significant. To control for multiple testing among estimated monthly or quarterly deviations for each diagnosis entity, we calculated false discovery rate (FDR)-adjusted $p$-values for all monthly and quarterly deviations using the Benjamini-Hochberg method, with the FDR threshold set at 0.05[29]. Data extraction was performed using Amazon Redshift version 1.0.38698 (PostgreSQL 8.0.2), and all statistical analyses and visualizations[30] were produced using R version 4.1.0[31,32]. To reduce concerns regarding our choice of a maximum allowable RMSPE, or the criteria we applied to identify diagnosis records and patients eligible for inclusion, we conducted sensitivity analyses to demonstrate no major changes in primary outcomes across different model performance thresholds or sets of inclusion criteria (Supplementary Note 3).

**Reporting summary.** Further information on research design is available in the Nature Portfolio Reporting Summary linked to this article.

## Results

EHRs from 44.62 million unique patients and 2455 US ophthalmic practices are included in this study, with a mean (SD) out-of-sample RMSPE of 10.3% (7.3%) among the predictive models used to establish counterfactual expectations for the 261 diagnosis entities meeting study criteria. Sustained reductions in care utilization are seen across nearly all diagnoses, including conditions that represent the least severe forms of the leading

causes of treatable low vision and blindness in the US[28] (Fig. 2): early-stage dry AMD, non-proliferative DR without diabetic macular edema (DME), age-related cataract, and glaucoma suspect. Across all conditions, the sharpest decreases in utilization occur during the hiatus (from March to May 2020, with a nadir in April 2020); and despite a rebound, utilization mostly (for 94.3% of diagnoses) remain below pre-pandemic volumes in the post-hiatus period (Fig. 3a). On average, deviations are below expectation by 67% (14%) in the nadir of the hiatus ($\delta_H = -0.67$), and by 13% (9%) post-hiatus ($\delta_{PH} = -0.13$) (Fig. 3b).

**Care utilization patterns in relation to disease severity.** Despite an overall reduction in care utilization, decreases tend to be smaller for more severe conditions, suggesting a continued prioritization of care for diagnoses perceived as more urgent. This inverse relationship between condition severity and the magnitude of underutilization is first seen among the 13 diagnosis categories encompassing all 261 diagnosis entities (Fig. 3b). Throughout the pandemic study period, utilization reductions were less pronounced for the more severe diagnostic categories of ocular globe injuries/intraocular foreign bodies (OGI/IOFB) (mean within-category deviations: $\delta_H = -0.49$, $\delta_{PH} = -0.08$), uveitis and ocular inflammation ($-0.51$, $-0.12$), and retina and vitreous conditions ($-0.60$, $-0.11$), whereas the less severe categories of refractive error ($-0.89$, $-0.14$), strabismus ($-0.84$, $-0.15$), and blindness and vision defects ($-0.78$, $-0.17$) consistently experience greater decreases in utilization.

Associations between severity and utilization are also evident among diagnosis entities themselves, but this trend is not consistent throughout different pandemic phases. During the hiatus, differences in utilization levels based on disease severity are conspicuous among nearly all diagnosis entities but these differences become less apparent in the post-hiatus period for some groupings of conditions. Among the common eye conditions of DR, AMD, glaucoma, and cataract, deviations from expected utilization are clearly separated by condition severity in April 2020, but not during the post-hiatus period (Fig. 4a). For instance, more severe stages of AMD show smaller reductions in visits during the hiatus ($\delta_H = -0.32$, $-0.53$, $-0.75$, $-0.81$ for wet AMD with active choroidal neovascularization (CNV); wet AMD with inactive CNV; early-stage dry AMD; drusen of macula, respectively); however, post-hiatus, the differences among stages are minimal ($\delta_{PH} = -0.16$, $-0.10$, $-0.15$, $-0.17$). On the other hand, the relative rankings of these deviations remain consistent over time for other sets of conditions like those related to neuro-ophthalmic diseases, as reflected by a strong positive correlation between hiatus and post-hiatus deviations ($r = 0.73$, $p = 0.001$) (Fig. 4b). Care utilization for oculomotor ($\delta_H = -0.52$, $\delta_{PH} = -0.07$) and abducens ($-0.59$, $-0.07$) nerve palsies, and optic neuritis ($-0.63$, $-0.06$) exhibit relatively limited decreases in visits throughout both pandemic sub-periods, whereas pupillary function anomalies ($-0.84$, $-0.19$) and irregular eye movements ($-0.83$, $-0.16$) consistently show greater utilization reductions. The presence of a strong positive correlation between hiatus and post-hiatus deviations suggests that conditions with greater reductions in care utilization during the hiatus period continue to exhibit relatively lesser rebounds in utilization post-hiatus. This could indicate that utilization for some categories of conditions continued to be sensitive to pandemic-related constraints to healthcare provision more so than other sets of diagnoses in the post-hiatus phase, but further research is needed to form a robust interpretation. Discrepancies in utilization levels based on condition severity, and relationships between hiatus and post-hiatus deviations, can be similarly observed among conditions in other diagnosis categories (Supplementary Figs. 1–6).

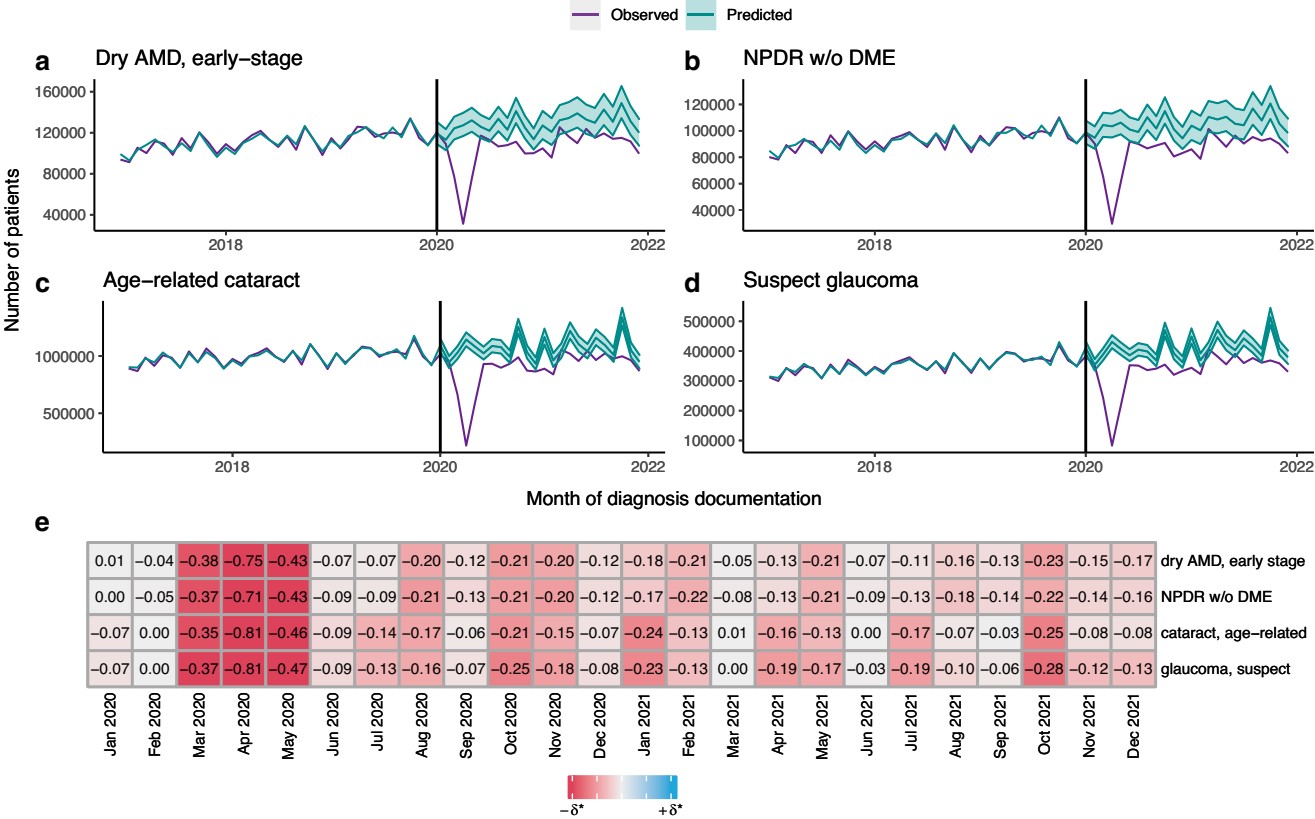

**Fig. 2 Deviations from expectation for common non-severe eye conditions.** Time series of observed (purple line) and predicted (teal line for mean predictions; teal shading for 95% prediction intervals) monthly numbers of patients documented with four eye conditions that represent the least severe forms of the leading causes of low vision and blindness: **a** early-stage dry age-related macular degeneration (dry AMD, early-stage), (**b**) non-proliferative diabetic retinopathy without diabetic macular edema (NDPR w/o DME), (**c**) age-related cataract, and (**d**) suspect glaucoma, from January 2017 to December 2021. The black vertical line at January 2020 denotes the start of the pandemic study period, the period for which monthly deviations from expectation are computed (**e**). A diverging color scale in the heatmap (**e**) is used to illustrate the direction, magnitude, and statistical significance of each monthly deviation, with the darkness of each cell a function of the product between the magnitude of the deviation and the negative log of its adjusted *p*-value. *Abbreviations*: AMD age-related macular degeneration, NPDR non-proliferative diabetic retinopathy, DME diabetic macular edema, w/o without.

Furthermore, among the 36 ocular emergencies with an associated severity ranking, median deviations from expectation progressively increase with condition severities during both hiatus and post-hiatus periods (Fig. 5a). For every unit increase in severity ranking, deviations increase on average by 5.5% ($p < 0.001$) during the hiatus, and 1.9% ($p = 0.04$) post-hiatus. Among sets of AMD, DR, and glaucoma diagnoses, NVT conditions consistently exhibit greater reductions than their VT counterparts during April 2020 ($p = 0.06, 0.03, 0.01$, respectively), but this relationship is non-existent or weaker post-hiatus ($p = 0.56, 0.89, 0.09$) (Fig. 5b).

**Identification of clusters of diagnosis entities with similar deviation patterns over time.** We summarize patterns of deviations in care utilization over time across all diagnoses using a cluster heatmap of quarterly post-hiatus deviations (Fig. 6), juxtaposed with April 2020 deviations, model performance errors, and time-to-recovery. We identify 33 conditions that experience the most intense utilization reductions in the post-hiatus phase, defined as having an average monthly decrease in utilization of 20% or more over this period that is statistically significant (i.e., $\delta_{PH} \leq -0.20$ with $p \leq 0.05$; also represented by dark shades of red in the circular heatmap of Fig. 6); many of these conditions are asymptomatic, slowly progressing, and/or NVT (Table S5A). The

diagnosis categories most represented in this set of conditions with the largest post-hiatus utilization reductions are cornea and external diseases (e.g., conjunctivitis-related diagnoses, peripheral corneal degeneration), followed by retina and vitreous conditions (e.g., retinal microaneurysms, unspecified background retinopathy, venous engorgement, and other retinal microvascular abnormalities), oculofacial plastics and orbital conditions (e.g., in situ carcinoma of the eye, benign eyelid neoplasm, orbital floor fracture, and other eyelid degenerative disorders), and blindness and vision defects (e.g., visual loss, suspect amblyopia, and color vision deficiencies) (Supplementary Table 3A). Conjunctivitis-related diagnoses are particularly well-represented among the set of conditions that exhibited intense post-hiatus utilization reductions, with presentations for infectious keratoconjunctivitis decreasing the most ($\delta_{PH} = -0.38$, 95% CI: $-0.41$ to $-0.35$, $p < 0.001$) among all diagnosis entities. No conditions that had a mean post-hiatus utilization reduction of 20% or more also exhibit partial or full recovery, except for the diagnosis of eyelid/periocular superficial injury ($\delta_{PH} = -0.21$, 95% CI: $-0.24$ to $-0.17$, $p < 0.001$), which experiences a partial recovery in November 2020.

Few conditions ($15/261 = 6\%$) meet or exceed counterfactual utilization predictions in the post-hiatus period (i.e., $\delta_{PH} \geq 0$) (Supplementary Table 3B); but among those that do, many are retinal and/or pediatric diseases, like unspecified DR with

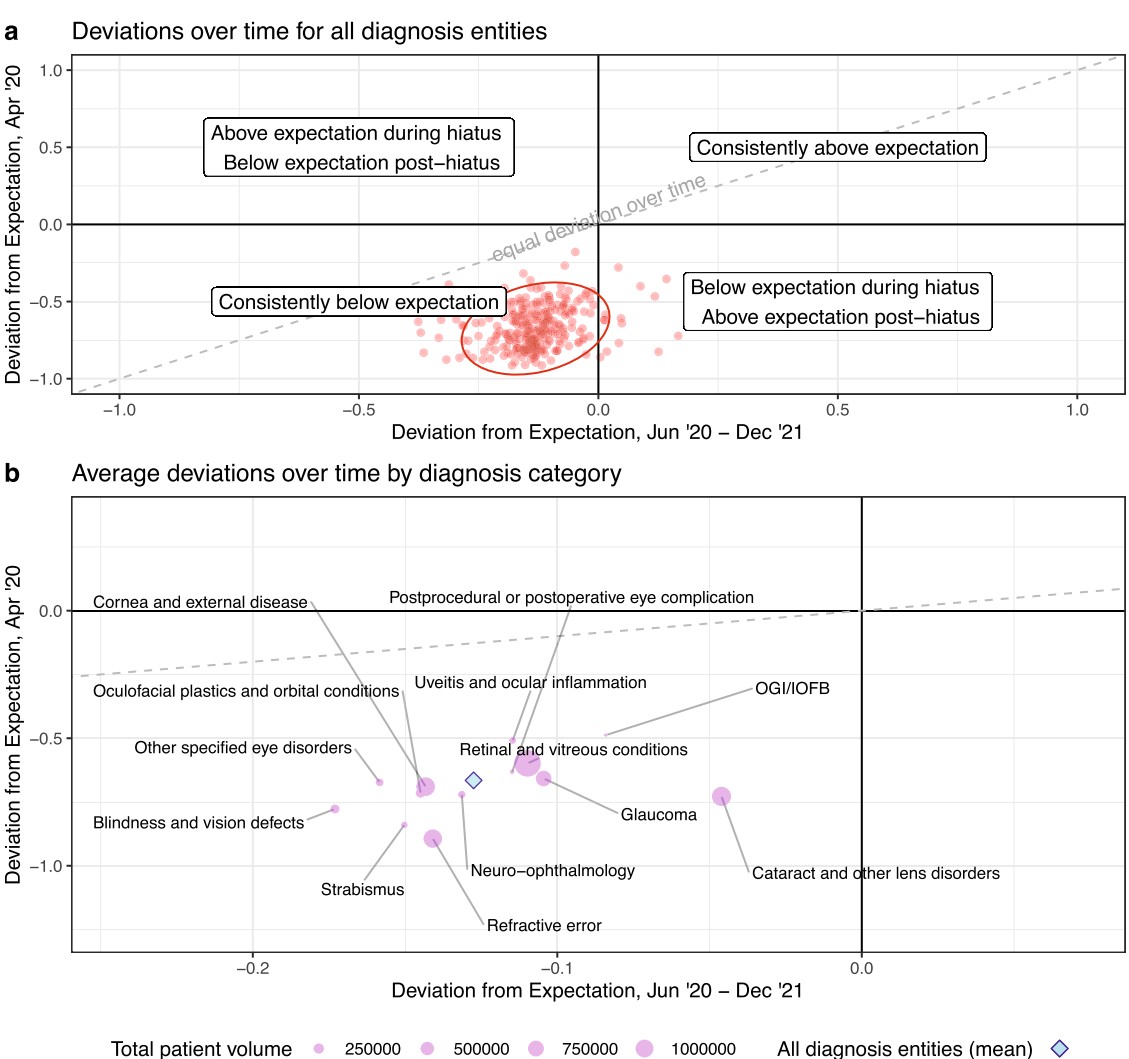

**Fig. 3 Hiatus vs. post-hiatus deviations across all diagnosis entities.** Deviations during the nadir of the hiatus period (April 2020) are plotted against deviations in the post-hiatus period (June 2020– December 2021), for all 261 diagnosis entities individually (red points in (**a**)) and averaged across the diagnosis entities that belong to each diagnosis category (pink circles in (**b**)). In (**a**), the 95% normal data ellipse (red oval) represents an estimated probability contour that is expected to contain 95% of all plotted diagnosis entities, and a line of equality (dashed gray line) represents no change in deviations over time (i.e., deviations during April 2020 are equal to post-hiatus deviations). In (**b**), the size of each point corresponds to the cumulative number of average monthly patients for the diagnosis entities within a category. The blue diamond represents the average of all deviations for April 2020 (−0.67, standard deviation (SD): 0.14) and the post-hiatus period (−0.13, SD: 0.09) across all 261 diagnosis entities. *Abbreviations*: OGI/IOFB ocular globe injury/intraocular foreign bodies, Postop complications postprocedural or postoperative eye complication, Oculoplastics and orbital conditions oculofacial plastics and orbital conditions, SD standard deviation.

($\delta_{PH} = 0.46$, 95% CI: 0.37 to 0.55, $p < 0.001$) and without ($\delta_{PH} = 0.04$, 95% CI: −0.01 to 0.09, $p = 0.11$) DME, infantile/juvenile cataract ($\delta_{PH} = 0.17$, 95% CI: 0.12 to 0.21, $p < 0.001$), eye injuries such as corrosion of the cornea/conjunctival sac ($\delta_{PH} = 0.14$, 95% CI: 0.08 to 0.21, $p < 0.001$) and ocular laceration without prolapse ($\delta_{PH} = 0.09$, 95% CI: 0.04 to 0.14, $p < 0.001$), and various stages of ROP: ROP stage 3 ($\delta_{PH} = 0.12$, 95% CI: 0.06 to 0.18, $p < 0.001$), ROP stage 2 ($\delta_{PH} = 0.04$, 95% CI: −0.02 to 0.11, $p = 0.17$), and ROP with unspecified stage ($\delta_{PH} = 0.05$, 95% CI: 0.00 to 0.10, $p = 0.07$). All 15 diagnosis entities that meet or exceed post-hiatus counterfactual utilization levels also experience recovery, with most of these conditions (12/15 = 80%) fully recovering.

Among all diagnosis entities, a broader set of conditions (116/261 = 44%) exhibit some form of recovery (Supplementary Table 4); however, many of these recoveries are not sustained (66/116 = 57%). The diagnosis categories with the highest proportions of conditions experiencing recoveries in utilization are uveitis and ocular inflammation (12/15 = 80%), postprocedural or postoperative eye complications (4/5 = 80%), ocular globe injuries/intraocular foreign bodies (3/4 = 75%), and cataract and other lens disorders (5/7 = 71%) (Fig. 6). On the other hand, the diagnosis categories with the lowest proportions of conditions experiencing recovery are other specified eye disorders (2/13 = 15%), followed by refractive error (2/7 = 29%), retina and vitreous conditions (28/75 = 37%), blindness and vision defects (5/13 = 38%), and cornea and external disease (22/56 = 39%) (Fig. 6). Approximately half of all conditions in the diagnosis categories of oculofacial plastics and orbital conditions (15/29 = 51.7%), neuro-ophthalmology (8/16 = 50%), strabismus (3/6 = 50%), and glaucoma (7/15 = 47%) experience recovery (Fig. 6). Among all diagnosis entities that exhibit partial or full recovery, the most common month for recovery to occur is June 2020 (42/116 = 36%), followed by September 2020

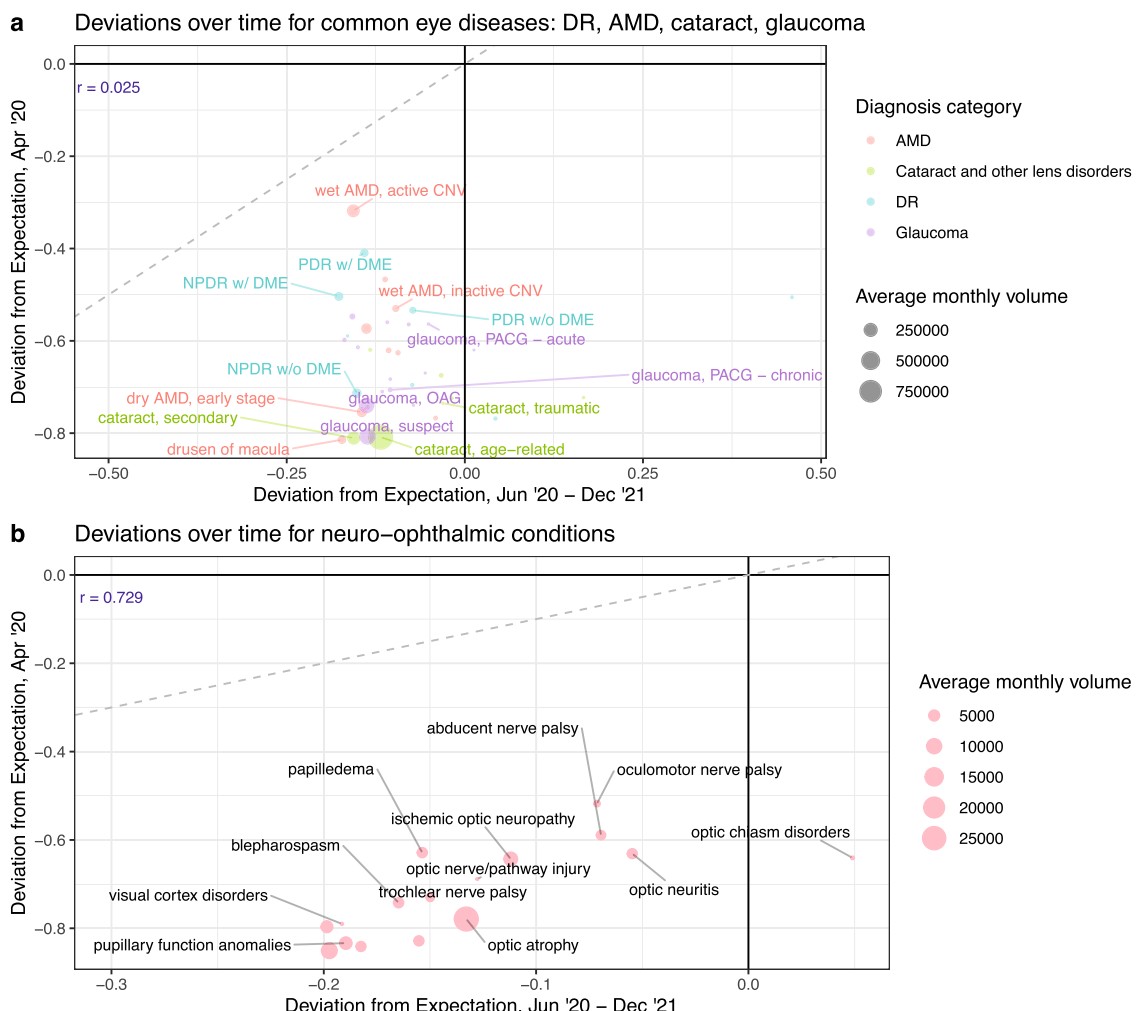

**Fig. 4 Hiatus vs. post-hiatus deviations for selected ophthalmic conditions.** Deviations during the nadir of the hiatus period (April 2020) are plotted against deviations in the post-hiatus period (June 2020–December 2021), for diagnosis entities that correspond to common eye diseases (**a**), and those that belong to the neuro-ophthalmic disease category (**b**). The size of each point corresponds to the average number of monthly patients for the diagnosis entity. Points were selectively labeled based on relevance (e.g., we excluded labels of diagnosis entities with names that contain key words such as *other* or *unspecified*). A Pearson's product moment correlation coefficient (r) is reported for the distribution of points in each plot (blue text; upper left corner), and a line of equality (dashed gray line) represents no change in deviations over time. *Abbreviations*: DR diabetic retinopathy, AMD age-related macular degeneration, CNV choroidal neovascularization, PDR proliferative diabetic retinopathy, NPDR non-proliferative diabetic retinopathy, DME diabetic macular edema, PACG primary angle-closure glaucoma, OAG open-angle glaucoma, w with, w/o without.

(17/116 = 15%), December 2020 (14/116 = 12%), June 2021 (14/116 = 12%), and February 2021 (13/116 = 11%) (Fig. 6).

Monthly deviations for all diagnoses, along with 95% CIs and p-values (unadjusted and adjusted), are provided in the supplement (Supplementary Fig. 7, Supplementary Data 3).

## Discussion

Our study presents a comprehensive exploratory analysis of care utilization patterns for ophthalmic diagnoses during the first two years of the COVID-19 pandemic in the United States. Although the prioritization of care for more severe conditions in the early pandemic phase has been previously reported across other specialties[33] and among select ophthalmology practices[34–36], this study contains distinct advantages over previous research. First, unlike prior investigations that only focused on broad diagnostic categories or a limited range of clinical problems, the wide spectrum of 261 granular ophthalmic conditions included in this analysis provides both an expansive and detailed view into the evolving visit patterns for diagnoses spanning a single specialty. Second, our analysis includes multiple

subperiods of the pandemic extending beyond its acute phase. This broad temporal scope facilitates versatile explorations of utilization patterns over time, including comparisons of how deviations from expected utilization levels varied over different subperiods for the same diagnosis entity or category, and measurements of the time it took for conditions to reach or exceed these counterfactual expectations. Another key strength of our study lies in the usage of predictive models trained on multiple years of pre-pandemic data to establish counterfactual utilization levels for each diagnosis entity, which provides a robust baseline for inferring the effect of the pandemic on care utilization. Furthermore, this investigation also examines how changes in care utilization during the pandemic may be related to attributes of diseases themselves, offering insights into characteristics associated with conditions that may have been prioritized during the pandemic. Collectively, the contributions of our analysis address several key gaps in the existing literature of pandemic utilization studies[33] and help advance a more thorough understanding of pandemic-driven shifts in care utilization for ocular conditions.

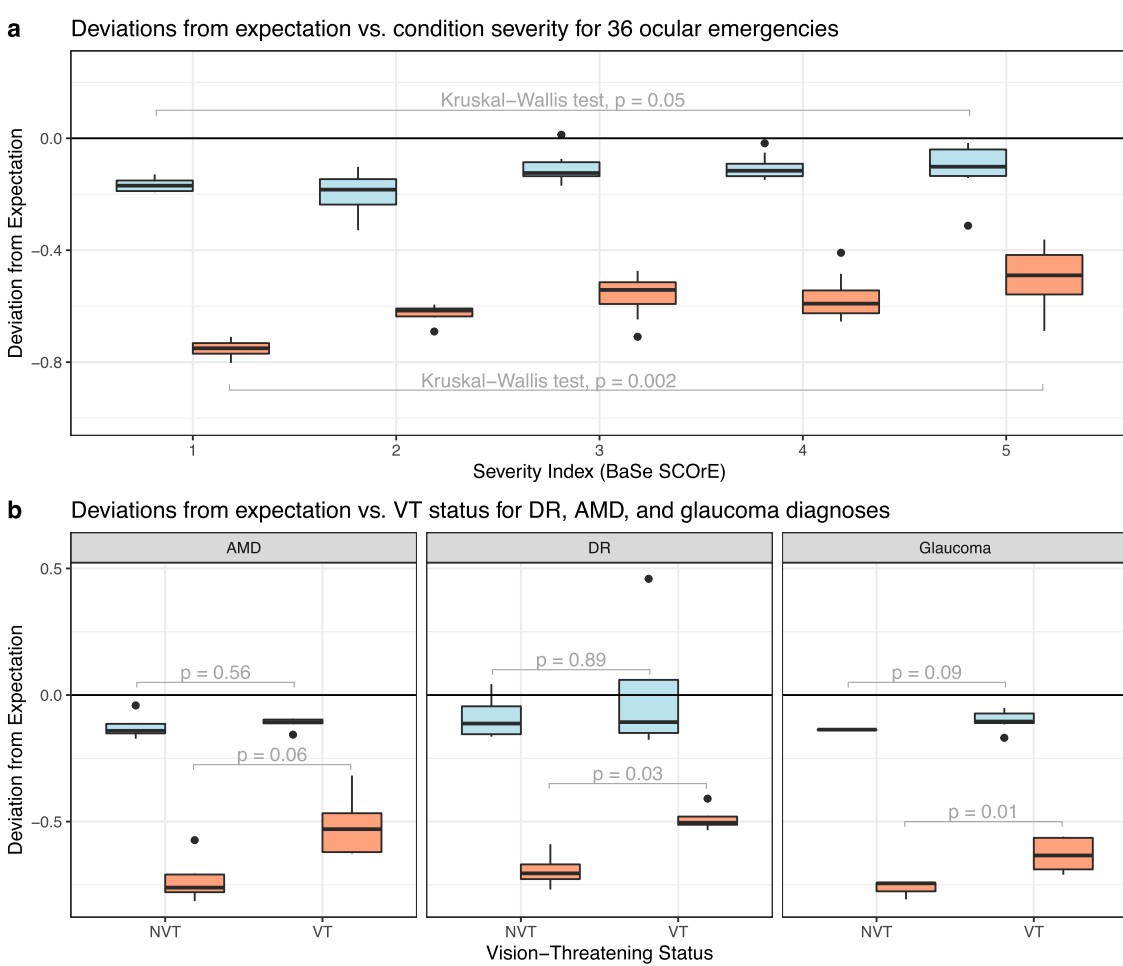

**Fig. 5 Deviations by severity level for ophthalmic emergencies and common conditions.** Boxplots depicting distributions of deviations from expectation, during the hiatus (orange boxplots) and post-hiatus (blue boxplots) periods, stratified by (**a**) increasing levels of severity (derived from the aggregate BaSe SCOrE's compiled by Bourges et al.) for a set of $n = 36$ diagnosis entities considered as common ocular emergencies, and (**b**) vision-threatening (VT) vs. non-vision-threatening (NVT) status for age-related macular degeneration (AMD), diabetic retinopathy (DR), and glaucoma diagnoses (Table S4) ($n = 28$ diagnosis entities in total). Outliers, indicated as black dots, are data points that are located at a distance greater than 1.5 times the interquartile range from either the lower quartile or the upper quartile of the boxplot. The 'whiskers' of the boxplots, which extend from the boxes as black vertical lines, represent the range of values that lie within 1.5 times the interquartile range from the lower and upper quartiles. To test for statistically significant differences in the central tendencies between distributions of deviations, we used Kruskal-Wallis (**a**) and Mann-Whitney U (**b**) tests to compute *p*-values (gray text). *Abbreviations*: BaSe SCOrE BAsic SEverity Score for Common OculaR Emergencies, DR diabetic retinopathy, AMD age-related macular degeneration, VT vision-threatening, NVT non-vision-threatening.

We observe an inverse relationship between a condition's severity and its magnitude of underutilization during the pandemic, but there is heterogeneity in the strength and endurance of this relationship across different sets of diagnosis entities. This invites investigation into other factors to explain why some conditions may have been prioritized over others. For instance, exploring associations between pandemic utilization levels and other attributes of diagnosis entities, such as the degree of required follow-up clinical visits and treatment, could provide additional insights. Furthermore, the lack of a clear separation between the post-hiatus deviations of many diagnoses in our study could reflect a general reduction in the willingness of patient populations to seek care (due to, e.g., heightened patient risk perceptions, economic hardships, or shifts in habits) rather than resource limitations that constrained the provision of care. Older adults, which comprise a substantial proportion of the patient population seen in ophthalmology, can be particularly susceptible to the avoidance of medical care. In a 2021 survey of

18,000 older adults from 11 high-income countries, American senior adults were found to be most likely to experience economic difficulties related to the pandemic; and among seniors with two or more chronic conditions, those in the US reported postponements or cancellations of appointments most frequently[37]. Similarly, in the Netherlands, older adults with multiple chronic conditions were also more likely to avoid medical care[38]. Furthermore, older Black and Latino/Hispanic American adults were found to be substantially more likely to experience economic hardships than older white American adults[37]. These findings highlight the need to better understand the complex effects of the pandemic on the healthcare utilization patterns of vulnerable populations, and to develop targeted strategies for equitable access to care.

Throughout the pandemic study period, there is a consistent decrease in visits related to the leading causes of blindness and vision loss among adults in the United States[28] and globally[39]: age-related macular degeneration, cataract, diabetic retinopathy,

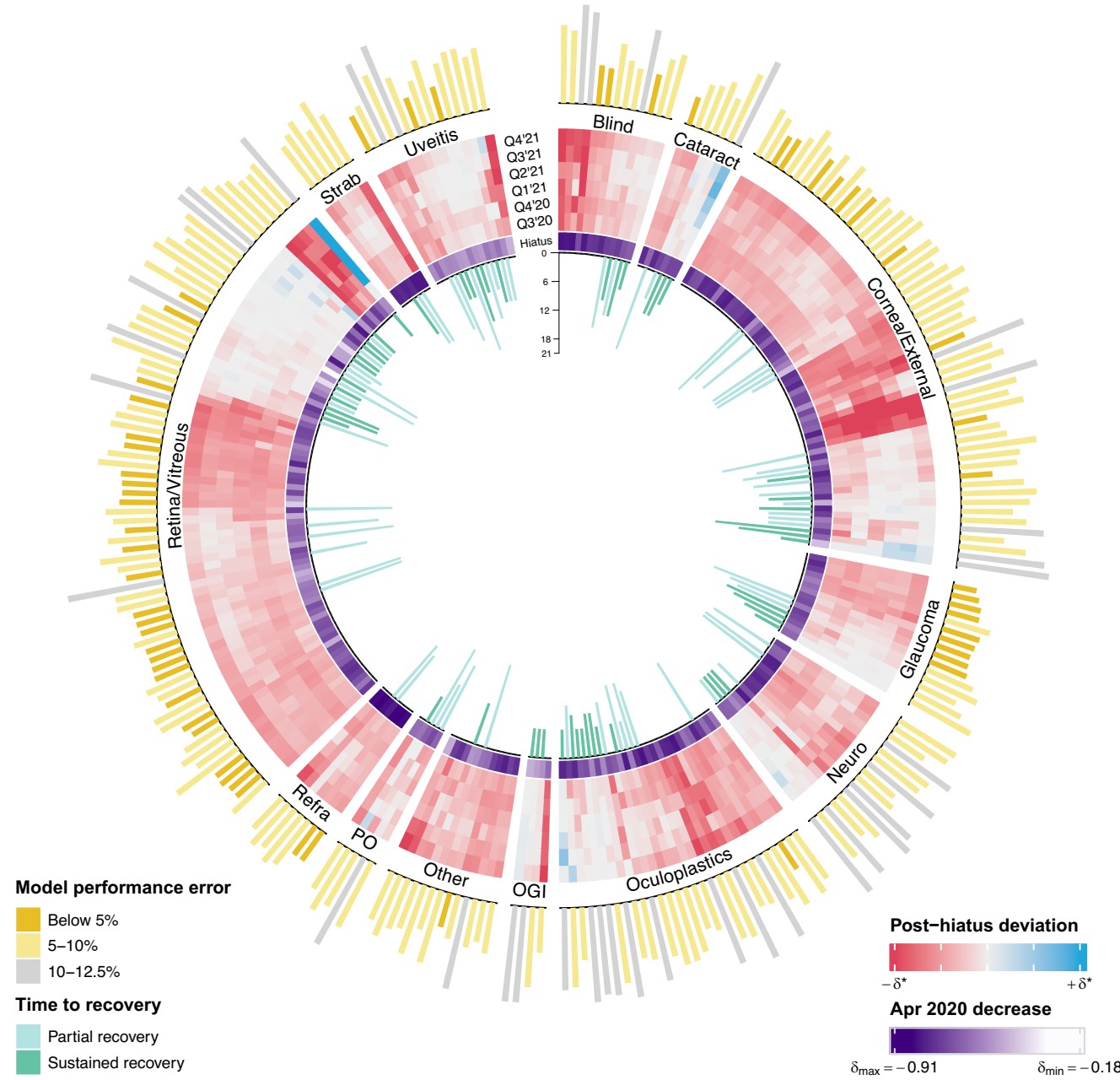

**Fig. 6 High-dimensional characterization of pandemic care utilization patterns.** Patterns of pandemic care utilization over time are illustrated for all 261 diagnosis entities, partitioned by 13 diagnosis categories. Beginning with the innermost (first) ring, barplots represent the number of months, starting from January 2020, it took for each diagnosis entity to achieve partial or sustained recovery of patient volumes. The absence of a barplot denotes no recovery. The second ring depicts the raw values of the deviations for each condition during April 2020 (sequential color scale). In the third ring, a cluster heatmap shows diagnoses' aggregate deviations for each quarter in the post-hiatus period, from the third quarter (Q3) of 2020 (innermost track of the heatmap) to the fourth quarter (Q4) of 2021 (outermost track), with the darkness of each heatmap cell a function of the product between the magnitude of the estimated quarterly deviation and the negative log of its adjusted p-value (diverging color scale). Diagnosis entities are clustered to group together conditions that exhibited similar patterns of post-hiatus deviations. In the outermost (fourth) ring, the magnitudes of each diagnosis entity's counterfactual model performance error (RMSPE) are visualized as barplots. *Abbreviations*: Blind blindness and vision defects, Cataract cataract and other lens disorders, Cornea/External cornea and external disease, Neuro neuro-ophthalmology, Oculoplastics oculofacial plastics and orbital conditions, OGI ocular globe injury/intraocular foreign bodies, Other other specified eye disorders, PO postprocedural or postoperative eye complication, Refra refractive error, Retina/Vitreous retina and vitreous conditions, Strab strabismus, Uveitis uveitis and ocular inflammation, RMSPE root mean squared percentage error, Q1 first quarter, Q2 second quarter, Q3 third quarter, Q4 fourth quarter.

and glaucoma. The early detection and timely treatment of these conditions are largely cost-effective and efficacious at restoring, or preventing further deterioration of, vision[39]. The burdens of these conditions are projected to grow due to ageing of the population (AMD, cataract, glaucoma) and global increases in diabetes

prevalence (DR)[39], underscoring the importance of considering the potential population health repercussions of these observed visit reductions.

Age-related macular degeneration, which has dry (non-neovascular) and wet (neovascular) forms, is the most common cause

of blindness in developed countries and is estimated to affect over 196 million people globally[40], and over 20 million people in the US[41]. Although most (nearly 90%) of vision loss from AMD is attributable to its neovascular form, which is less common than non-neovascular AMD, the sustained decreases in visits we observe for both forms of AMD diagnoses are concerning. Repeated intravitreal injections of anti-vascular endothelial growth factor (anti-VEGF) agents are needed to stabilize vision for patients with neovascular AMD, and it is also important for individuals at risk for progression to more advanced stages of AMD (e.g., patients with early-stage non-neovascular AMD and/ or a family history of AMD) to have regular dilated eye examinations to allow for adequate monitoring of disease progression and early detection of intermediate or advanced disease[42].

Cataracts rank as the top cause of blindness globally, particularly in low- and middle-income countries that face considerable barriers in access to cataract surgery. Worldwide, over 95 million people are affected by cataracts, with over 30 million cases in the US alone[28,43]. Due to the slowly-progressing nature of age-related cataracts, which are the most common form of cataract, the urgency of cataract detection and treatment is generally not quite as critical as for the other leading causes of blindness and vision loss. However, the diagnosis and monitoring of cataracts are still necessary to determine whether the presence of a cataract is contributing to visual impairment, which can inform treatment choices, and to assess whether the untreated cataract may be causing other vision-related problems[44,45]. To restore vision, visually important cataracts are treated with cataract surgery, a highly cost-effective intervention[39]. Despite its wide availability in the US[28], barriers in access to cataract surgery appear to exist, and there can be notable sociodemographic disparities in cataract prevalence and treatment outcomes[46]. Further research, such as studies examining cataract surgery volumes, may provide deeper insight into the long-term significance of the prolonged decreases in cataract diagnoses we are observing in this study.

Diabetic retinopathy, which is present in roughly a third of people with diabetes, is the most common cause of blindness among working aged adults (e.g., ages 20–74) globally[39] and in the United States[28]. With over 150 million people affected worldwide[47] and 9.6 million in the US[48], DR is the only leading cause of blindness and visual impairment that is experiencing an increase in age-standardized prevalence[39]. In the absence of appropriate intervention, DR progresses from mild to more severe stages; but with early detection and timely treatment, 90-95% of blindness caused by vision-threatening DR can be prevented[39,49]. Management strategies for DR, including the recommended follow-up frequency (ranging from once every 12 months to more frequently for more rapidly progressing disease) and treatment approaches (such as laser surgery treatments and intravitreal anti-VEGF injections, among others), are tailored based on the severity of the retinopathy as well as the presence and vision-threatening status of DME, which is the accumulation of fluid in the macula due to leaky blood vessels[49]. Because it is common for patients with DR to remain asymptomatic for years, even at some more advanced stages, annual screenings via dilated eye exams are recommended for individuals with diabetes; however, the actual rates of screenings and ophthalmic care referrals have not met guideline recommendations[28,49]. Furthermore, for certain forms of DR, such as high-risk proliferative DR (PDR), prompt treatment is required[49]. Thus, the persistent declines in visits we are finding for diagnoses of both non-proliferative DR (NPDR) and PDR, with or without DME, may suggest a looming challenge in the effective management of this condition at the population level.

Glaucoma is a leading global cause of irreversible blindness and visual morbidity, estimated to affect more than 76 million people

globally[50] and over 3 million in the US[51]. Despite its varied subtypes, which can be broadly divided into open-angle or angle-closure glaucoma, all forms of glaucoma share a common pathological feature: the degeneration of the optic nerve, frequently associated with elevated intraocular pressure (IOP)[52]. For chronic forms of glaucoma, this degeneration is progressive and asymptomatic until it leads to permanent vision loss, making its detection particularly challenging, with less than half of people with glaucoma aware that they have the disease[28,39,53]. Long-term management strategies, aimed at reducing IOP through topical medication, laser therapy, or incisional surgery, can be limited in success by factors such as patient non-adherence to medication dosing regimens and the transient effectiveness of treatments (e.g., laser trabeculoplasty) in providing sustained control of IOP[39,54,55]. Therefore, targeted screening strategies and regular monitoring are needed to identify and treat glaucoma earlier on in the course of disease, and the sustained visit reductions we are observing in this study for patients with glaucoma suspect and established diagnoses of both open-angle and angle-closure glaucoma may signify an emerging population vision health concern.

In addition to finding consistent declines in visits for diagnoses corresponding to the leading causes of visual impairment, we also identify clusters of diagnoses that experience exceptional deviations in utilization patterns over time by highlighting conditions with above- and below-average utilization changes and distinguishing the relatively few conditions experiencing recovery from those that do not. Conditions exhibiting intense reductions of post-hiatus utilization mostly include those that are non-vision-threatening (e.g., conjunctivitis) or are precursors of vision-threatening conditions (e.g., subclinical indications of DR like retinal microaneurysms/background retinopathy), whereas those with above-average utilization include ocular emergencies with a risk for irreversible vision loss such as ROP. Further research, such as customized cohort studies designed to investigate the long-term clinical impact of missed or delayed care for specific condition(s) of focus, is needed to better establish distinctions between benign reductions in visits from potentially harmful ones like decreases in screenings for common conditions that can lead to irreversible vision loss[56]. Additionally, developing condition-specific definitions of loss to follow-up (LTFU) based on clinical considerations such as disease pathology and the appropriate frequency of follow-up or treatment[57], enables targeted cohort studies to shed light on the patient-level determinants of LTFU during the COVID-19 pandemic.

This study has several limitations. Our parametric approach to establish counterfactual expectations of utilization introduces the possibility for model misspecification. Furthermore, we only use three years' worth of pre-pandemic data to train models because the analysis is restricted to the usage of only one diagnostic coding system (ICD-10). However, model selection from a variety of candidate algorithms using leave-out-one-year cross validation enhances the predictive ability of our counterfactual models, and even with three years of data, we achieve low RMSPE. In some cases, observed deviations from expectation may reflect artificial factors extraneous to true changes in utilization, such as adjustments in ICD-10 documentation practices and undetected data latency issues. Because we account for potential data latency issues by excluding practices that did not contiguously contribute data throughout the study period, our analysis is biased towards practices that were able to remain operational throughout the pandemic, despite sensitivity analyses demonstrating no major changes in primary outcomes. Additionally, due to current database limitations and the large volume of diagnosis entities in the analysis, we cannot directly identify the primary condition or disease at each patient visit, and cannot readily distinguish, at

scale, newly occurring incident diagnoses from ones that may be repeat documentations of historical conditions or recurrent/ongoing clinical problems. Although this study uses the largest and most comprehensive registry of eye diseases in the United States, our results may not capture potential shifts in ocular care volume that occurred during the pandemic from practices integrated with the IRIS Registry to eye care settings not covered by the database (e.g., at optometrists who are not employed by ophthalmologists, ophthalmic practices using paper charts, or some tertiary academic medical centers that are not included in the IRIS Registry). Finally, there are inherent ambiguities associated with terms like severity, which are also reflected in the external measures of disease severity we use for hypothesis testing.

Nonetheless, given the COVID-19 pandemic's lasting effects on the allocation of healthcare resources, the public health implications of widespread and prolonged disruptions to care are increasingly important to understand[10]. An exploratory but expansive characterization of care utilization patterns across a specialty may serve as an initial tool for researchers and clinicians to evaluate the extent, magnitude, and differential impact of these disruptions. Here, using data from the United States' first comprehensive ophthalmic registry, we employ a common analytical framework to identify patterns of patient presentations during the first two years of the COVID-19 pandemic across a broad set of granular ophthalmic conditions. This framework can be flexibly adapted to examine pandemic utilization trends among services from other specialties and across different healthcare settings. High-dimensional and high-resolution characterizations of care utilization may enable timely and ongoing assessments of the overall state of healthcare usage across a specialty, inform targeted efforts to investigate the pandemic's long-term impact on clinical outcomes and burdens of disease, and highlight factors that may be driving the differential utilization of care.

## Data availability

The original data that support the findings of this study are from the American Academy of Ophthalmology IRIS® Registry, which is not a publicly available dataset. However, the minimum data necessary to interpret, verify, and extend the research are currently available in the Supplementary Information, Supplementary Data, and at https://github.com/charlesli37/covid-oph-dx-utils. Source data for Figs. 2–6 of the main text are available as Supplementary Data 4. Eligible investigators may apply for research opportunities to work with IRIS Registry data; more information is available at https://www.aao.org/iris-registry/data-analysis/requirements or by contacting irisregistry@aao.org.

## Code availability

SQL and R programming scripts used for this study are available at https://github.com/charlesli37/covid-oph-dx-utils[58].

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

## Author contributions

C.L., F.L., E.M.C., P.A.C. and J.R.H. designed the study. F.L., E.M.C., R.N.K., E.T.C. and R.S.M. helped curate the definitions of diagnosis entities and their assignment to diagnosis categories. P.A.C. and J.R.H. verified the analytical methods. C.L. extracted and analyzed the data. F.L., D.W.P., and S.D.M. supervised the project and contributed to the interpretation of results. C.L. wrote the paper with input from all authors.

## Competing interests

The authors declare no competing interests.
