## [Peer Review File · Communications Medicine]

Reviewers' comments:

Reviewer #1 (Remarks to the Author):

Li and coauthors report on the utilization of ophthalmic care during the covid 19 pandemic using the IRIS database.

Abstract:

Please list the leading causes for vision impairment in the results.

Introduction:

I would agree that ophthalmology did experience one of the largest drops in volume, but for the most part, has returned to near pandemic levels. Many industry sites have this information freely available.

Methods:

While the use of the IRIS registry has tremendous upside, as the authors point out, it only accounts for 70% of ophthalmologists. How is this accounted for when studying multiple time periods. If a physician or practice join, for say in 2021, how did that affect the data pre-pandemic?

Additionally, to my knowledge, IRIS is not linked with EPIC or MDI, two extremely large EHR systems that house many academic and large practices, which may have been the primary drivers of ophthalmic care during the pandemic. This certainly may affect some of the metrics and learning done. Perhaps these conditions are missed from the model and thus are artificially deemed nonimportant, but when in fact, the conditions were still being evaluated at tertiary centers not in IRIS. I know this limitation may not be addressable, but perhaps providing a breakdown of the types of physicians in the database would give a context and perhaps more generalizability of the results.

How was the primary condition or disease identified? Many patients may have per se a retinal detachment, but it is old, but have new strabismus?

Results:

In regards to the change in utilization, perhaps patients are being sent elsewhere? Are optometrists captured in IRIS. Perhaps MDs are trying to outsource refractive care to ODs?

Discussion:

How does this data compare to other specialties generally treating older patients?

One study not cited by Xu and colleagues does in fact include years of pre-pandemic data on retina care utilization.

The last paragraph about the utility of this data. I suggest removing policy makers. This is a slippery slope in medicine, especially with what is being done to in obstetrics.

Overall, I think this is an interesting concept which may help guide directions of care in the future. I think the database used is not the best representation of ophthalmic care given the limitations and

practices included. Perhaps using on a more homogeneous population would give more accurate results for ophthalmology, but from a purely methodologic standpoint this paper does provide a foundation for future work.

Reviewer #2 (Remarks to the Author):

This is a very interesting study, using nation-wide eye care registry (AAO IRIS@ Registry) data involving 44.62 million patients and 2,455 practices, the authors investigated how the restriction of COVID-19 pandemic may affect the utilisation of ophthalmic care. Despite the fact that COVID-19 restriction has been eased in majority of the countries in the world, the study findings are far-reaching because it provides evidence to demonstrate how restriction of infectious disease or lock down may cause substantial disruption of care in the context of ophthalmic care, a specialist care with high elasticity of demand. This massive dataset allows a meaningful investigation on eye care utilisation across a board range of diagnoses (3400 ophthalmic ICD-10 diagnosis) using longitudinal electronic health record data from a large national disease registry.

The primary outcome of the study was deviation from expected care utilisation for each diagnosis entity calculated as relative difference between observed and expected numbers of patients for a given month. It is understandable that an accurate estimation on the expected number of patients is the key for the estimation on deviation because the observed number of patients during pandemic should be easy to observe. In the manuscript, it is difficult to understand how these expected numbers of patient for a given month were calculated from the description in Page 8 / Line 173-179. Page 7 / Line 155 to 171 explained how the counterfactual models ("step 2") were used to generate a linear trend over years, but it is hard to understand how the expected numbers of patients were calculated before the pandemic? Also how does the coverage of IRIS registry (I assume the coverage was improving over time) may contribute to the bias on estimating the expected numbers of service?

The secondary outcome of the study was to examine the time used to recover to expected values and further classified into sustained recovery and partial recovery., but this part of results was not described in sufficient details in the manuscript.

Elasticities of disease diagnosis categories: the authors described the diseases as more elastic if greater negative deviations were observed during the pandemic and in fact observed greater decrease among the disease with less severe conditions. As expected, they observed the emergency eye diseases such as ocular globe injuries or intraocular foreign body as the least elastic diagnosis whereas the less severe eye diseases, such as refractive error, strabismus were more elastic.

Line 273-280 the authors described 36 ocular emergencies and their deviation during pandemic, hiatus or post-hiatus period, in the context of severity ranking. This is interesting also. Something missing in the analysis of deviation is about the eye diseases that require continuous or regular follow-up and care, for example, glaucoma or anti-VEGF AMD patients would require regular follow-up visit and treatment, it would be good to observe how elasticity may be associated with the diseases at different level of requirement on regular follow up and treatment.

Reviewer #3 (Remarks to the Author):

Reviewer comments on Li et al 2022 Nature submission “Elasticities of Care Utilization Patterns During the COVID-19 Pandemic”

1. This paper is a clearly written and carefully documented discussion of the pattern of services for ophthalmic conditions in the US from 2017 through 2021. The study documents both the empirical magnitudes of reductions and the variation around expected trend values during the immediate COVID-19 hiatus period (April 2020) and the post-hiatus recovery period. The authors call the percentage reduction in spending during each period the “elasticity of care” and show that some diagnoses were much more elastic (i.e., reduced much more) than others, and in other cases they were much less so.

2. I have two main concerns about the paper, one about the paper’s clarity and the other about whether the paper answers the all-important “so what” question.

3. The authors use the term elasticities throughout the paper text without clearly defining what it is intended to measure. The concept of elasticities is used by economists, physicists and others to measure responsiveness with respect to some parameter or variable change, such as changes in income, price, or shock. I am unused to speaking of the elasticity of a variable’s response to an event, such as the pandemic, and found this usage in the paper confusing.

4. It is notable that the abstract does not mention the word elasticity, even though it is in the paper title. I find the text clear at line 47 “Less severe conditions experienced greater utilization reductions ...” and at line 49 they speak of “intense utilization reductions”, rather than using the term elasticities. The paper title and text could be revised to be clearer on this.

5. My second concern is that it is unsurprising that there is heterogeneity in responsiveness of specific diagnoses to COVID-19 across the over 300 diagnosis entities examined. The authors admit this in their discussion section where they write

Lines 309-313: “While the presence of an elasticity gradient that corresponds to condition severity may be unsurprising (particularly during the hiatus), the heterogenous strength and endurance of such a gradient across different sets of diagnoses invites future research to elucidate additional reasons to explain why some conditions may have been prioritized over others.”

The authors also note:

326 “Further research is needed to understand the distinction between
327 benign reductions in visits from potentially harmful ones...”

6. While I agree that the paper has shown an association between both the severity and vision threatening/non-vision threatening diagnoses and their magnitudes of hiatus and post-hiatus reduction in frequency, the paper does not attempt to demonstrate the longer term significance of this variation. Are there more patients developing more serious conditions, (e.g., cancers, loss of vision, or blindness) because of the reduction in diagnosis? This is all left to future research. This left me unsatisfied as a reader.

7. The central contribution of the paper would seem to be the multicolor Figure 6 which illustrates for all of the 261 diagnostic clusters seven different dimension of information about the type of the impact of COVID-19. I found this figure overly complex, confusing and not particularly informative, as the authors implicitly reveal in the fact that they discuss this complex figure for only five lines, at lines 283-287. Perhaps extracts from the extensive supplement tables on which it is based would be more informative.

8. After reading the supplement, I found table S5 more interesting, with the high and low outliers for post-hiatus changes.

9. I would have liked to have seen more discussion, perhaps in the supplement, about how the entities were merged from diagnoses. They are clearly much finer than the AHRQ CCSR. They represent a lot of work by someone. Perhaps in a new supplement text S5?

10. The paper should acknowledge online 124 the Agency for Healthcare Research and Quality Clinical Classification Software Revised (CCSR) version 2020.2.

11. The statistical methods used in the paper are of high quality and are clearly explained. The one issue I do not recall seeing discussed were corrections to t-statistics to account for the very large number of tests being conducted.

12. At line 178 the paper notes bounds on $\delta = (\text{OBS}-\text{EXP})/\text{EXP}$ as

Where $\delta \in [-1, 1]$

I do not see why +1 is the upper bound of δ , since the OBServed value could be more than twice EXpected during a recovery period. We are not talking about elastic bands here.

13. At line 229 the paper notes

229 On average, deviations were below expectation by 67%

230 (14%) in the nadir of the hiatus ($\delta_H = -0.67$), and by 13% (9%) post-hiatus ($\delta_{PH} = -0.13$) (Figure 231 3B).

While this may be a true statement, it is not shown in Figure 3B. Perhaps add a mention in that table note?

14. Supplement table S1 reveals that ten candidate prediction models were examined for predicting baseline. The most flexible Model 10 uses 18 parameters to predict group means for only 36 time periods. The modeling also allowed three adjustments for overfitting. This almost surely is overfitting the data. No sensitivity analysis is presented about this degree of overfitting, only the chosen specification for each diagnosis entity. The sensitivity analysis presented in Text S3 does not mention this concern.

15. It would make the tables more useful to viewers if the pdfs of the key table S2 included headers on each of its 100+ pages instead of only on the first page. This could use abbreviations for all variables so as to use only one line.) An Excel version could include the headers only at the top of the page.

16. Table S1 contains the full crosswalk of 336 diagnostic entities considered in this paper, along with the binary flag of the 261 actually included in the analysis. The flags appear to be assigned to specific ICD-10-CM codes but are actually applied to the diagnosis entities.

17. This paper uses 13 categories of eye disorders, as listed in table S7 at the end, but these groups are not given a name there. Twelve of these are identical to the 12 CCSR categories included in the AHRQ CCSR version 2020.2 as linked on the AHRQ website included in the references to the main paper. The only difference appears to be that the authors have added one further category OGI/IOFB category which combines certain eye injury codes S05.XXX that the CCSR includes with other types of injuries. Also in that group are other foreign body in eye codes. Given the extreme reliance on the AHRQ system, you should acknowledge this reliance on CCSR better in the manuscript than the current passing reference on line 124.

18. The mention of how the 336 diagnosis entities were created is inadequate. The text only includes the following:

124 All diagnosis

125 entities were required to have a sufficient utilization level (i.e., above single-digit patient counts)
126 for each month of the study period (January 2017 to December 2021).

.

This text should be expanded to indicate what information was made available to clinicians when grouping together individual diagnoses into your diagnostic entities.

19. Also imprecise or misleading is this statement

126 Conditions with poor

127 counterfactual model performance (considered as $\geq 12.5\%$ root-mean squared percentage error
128 [RMSPE]) were also excluded, resulting in a final set of 261 diagnosis entities attributed to 13
129 mutually exclusive diagnosis categories (e.g., blindness and vision defects, cataract and other
130 lens disorders, corneal and external conditions) (Table S1)

The number of entities excluded for the second reason (poor counterfactual model performance) appears to be those shown as having a RMSPE > 12.5 as shown in table S2, not S1. A flag on these entities in that table would be informative.

20. The index for the Supplement mentions but does not include the following on page 209.

References for the Supplementary Information Pg. 209

Reviewer Comments

Reviewer #1 (Remarks to the Author):

Li and coauthors report on the utilization of ophthalmic care during the covid 19 pandemic using the IRIS database.

It was a pleasure to consider and address the reviewer's thoughtful remarks and critiques of our research.

Abstract:

Please list the leading causes for vision impairment in the results.

We thank the reviewer for this suggestion. The leading causes of vision impairment have now been listed in the abstract, along with other changes to the abstract to satisfy word count limits:

L47-50: Analyzing records from 44.62 million patients and 2,455 practices, we **observed** lasting reductions in ophthalmic care utilization, including visits for leading causes of visual impairment (**age-related macular degeneration, diabetic retinopathy, cataract, glaucoma**).

Introduction:

I would agree that ophthalmology did experience one of the largest drops in volume, but for the most part, has returned to near pandemic levels. Many industry sites have this information freely available.

We appreciate this point and acknowledge that some analyses using industry data have reported recoveries in ophthalmic care utilization to pre-pandemic, or near-pre-pandemic, levels. For instance, using data from patient intake software (Phreesia) collected from 1,600 provider organizations in the United States, Mehrotra et al. reported that ophthalmology visits in December 2020 increased by 3% compared to the first week of March 2020.¹ However, the authors note that analytic and data limitations, such as a lack of adjustment for seasonal effects and a convenience sample of providers, may constrain the generalizability of their findings.

Furthermore, other sources have described sustained decreases in ophthalmic and overall healthcare utilization beyond the acute phase of the pandemic in Spring 2020. For example, a February 2022 survey administered by McKinsey & Co. to 101 large private-sector hospitals in the United States reported that outpatient ophthalmology visits decreased by 7% in January 2022 compared to January 2019, the second-largest decrease among all specialties included in the survey.² Using data from the CDC's National Health Interview Survey and the Quarterly Services Survey from the US Census, the Peterson-KFF Health System Tracker reported in January 2023 that about 1 in 4 American adults missed or delayed medical care in 2021 due to either the COVID-19 pandemic or healthcare costs, and that quarterly hospital discharge volumes up until the third quarter of 2022 still remained below 2018-2019 levels, in spite of a rebound since the second quarter of 2020.³

Considering these findings, the heterogenous ways of measuring healthcare utilization, and the results from our original analysis, we have not added any general statements about the recovery of ophthalmic or overall care utilization to pre-pandemic or near-pre-pandemic levels; but we would be glad to revisit this issue with additional editorial input.

Methods:

While the use of the IRIS registry has tremendous upside, as the authors point out, it only accounts for 70% of ophthalmologists. How is this accounted for when studying multiple time periods. If a physician or practice join, for say in 2021, how did that affect the data pre-pandemic?

We appreciate the need for clarification. For this study (as noted in L151-153 of the initial manuscript), we only considered data from practices that contributed records to the IRIS Registry database for each month of the global study period (2017-2021). The findings from this analysis should therefore not reflect changes in overall utilization trends that may be attributable to practices that newly opened or became permanently closed during the study period. Furthermore, our sensitivity analysis (Text S3) demonstrated no major changes in the primary outcome (estimated deviations from expected utilization levels), with or without requiring that the study be limited to data from practices that were consistently active for each month from 2017-2021. To improve clarity, we have edited the text:

L181-184: Furthermore, to exclude from consideration changes in utilization trends that may be attributable to new openings, permanent closures, or other changes in the data reporting statuses of practices, we only analyzed EHRs from practices that reported data to the IRIS Registry throughout all months of the global study period.

Additionally, to my knowledge, IRIS is not linked with EPIC or MDI, two extremely large EHR systems that house many academic and large practices, which may have been the primary drivers of ophthalmic care during the pandemic. This certainly may affect some of the metrics and learning done. Perhaps these conditions are missed from the model and thus are artificially deemed nonimportant, but when in fact, the conditions were still being evaluated at tertiary centers not in IRIS. I know this limitation may not be addressable, but perhaps providing a breakdown of the types of physicians in the database would give a context and perhaps more generalizability of the results.

We understand this concern. The IRIS Registry includes approximately 70% of active, practicing ophthalmologists, and is integrated with several EHR systems,⁴ including Epic and MDI. In fact, MDI is the top EHR system integration within the IRIS Registry. Although the penetration is not as large as with private practices, approximately 33% of Association of University Professors of Ophthalmology Member Institutions are also integrated with the IRIS Registry.

We have edited the text to include this information, and further note the inability of our study to capture possible shifts in ophthalmic care utilization that occurred during the pandemic to practices that are not covered by the IRIS Registry:

L166-169: Although its coverage is not as large as with private, outpatient practices, the IRIS Registry, which is compatible with multiple EHR software vendors, ^(Ref #19 in the main text) is also integrated with approximately 33% of Association of University Professors of Ophthalmology member institutions.

L455-460: Although this study used the largest and most comprehensive registry of eye diseases in the United States, our results may not capture potential shifts in ocular care volume that occurred during the pandemic from practices integrated with the IRIS Registry to eye care settings not covered by the database (e.g., at optometrists who are not employed by ophthalmologists, or tertiary academic medical centers that are not included in the IRIS Registry).

How was the primary condition or disease identified? Many patients may have per se a retinal detachment, but it is old, but have new strabismus?

Thank you for raising this point. We were unable to directly identify which diagnosis codes were designated as the primary condition/disease at each visit because the IRIS Registry database only contains information on dates of documentation for diagnosis codes (e.g., ICD-10-CM) corresponding to each patient or patient eye. Similarly, we were unable to distinguish whether diagnoses were new (i.e., incident) instead of recurring, chronic, or historical because we cannot reliably access direct information on the dates of resolution for diagnoses; and given the large number and diversity of diagnoses we studied, it was also not feasible to indirectly infer which conditions are newly diagnosed because doing so would require constructing individual patient cohorts with customized inclusion and exclusion criteria (e.g., no prior documentation of certain ICD codes within a given lookback period) for each of the 261 diagnosis entities that we examined. In the revision, we now describe these important limitations:

L451-455: Additionally, due to current database limitations and the large amount of diagnosis entities included for analysis, we were unable to directly identify the primary condition or disease at each patient visit, and could not readily distinguish, at scale, newly occurring incident diagnoses from ones that may be repeat documentations of historical conditions or recurrent/ongoing clinical problems.

Results:

In regards to the change in utilization, perhaps patients are being sent elsewhere? Are optometrists captured in IRIS. Perhaps MDs are trying to outsource refractive care to ODs?

Only care that is rendered by an optometrist who is employed by an ophthalmologist would be captured by the IRIS Registry. In the revision (*as noted above*), we now discuss this key limitation in **L455-460**.

Discussion:

How does this data compare to other specialties generally treating older patients?

We thank the reviewer for this question. While we were unable to locate studies from other specialties that were closely analogous to ours (e.g., ones that included data up until, or beyond, December 2021 on post-hiatus utilization trends for a granular and broad set of diagnoses or health services that span a single specialty), or find evidence to suggest that older patients consistently exhibited larger decreases in care utilization than younger patients,^{5,6} we have now included additional discussion on challenges in accessing healthcare that older adults (and American seniors in particular) were likely to encounter during the pandemic:

L410-421: Older adults, which comprise a substantial proportion of the patient population seen in ophthalmology, may have been particularly susceptible to the avoidance of medical care. In a 2021 survey of 18,000 older adults from 11 high-income countries, American seniors were found to be most likely to experience economic difficulties related to the pandemic; and among seniors with two or more chronic conditions, those in the US reported postponements or cancellations of appointments most frequently.⁷ (Ref #37 in the main text) Similarly, in the Netherlands, older adults with multiple chronic conditions were also more likely to avoid medical care.⁸ (Ref #38 in the main text) Furthermore, older Black and Latino/Hispanic American adults were found to be substantially more likely to experience economic hardships than older white American adults.⁷ (Ref #37 in the main text) These findings highlight the need to better understand the complex effects of the pandemic on the healthcare utilization patterns of vulnerable populations, and to develop targeted strategies to ensure equitable access to care.

One study not cited by Xu and colleagues does in fact include years of pre-pandemic data on retina care utilization.

Thank you for bringing this interesting study to our attention. We have amended our description on the relative advantages of our study to convey that, unlike many other studies on pandemic care utilization that reported deviations from expected utilization levels by directly calculating differences between observed patient volumes at various pandemic vs. pre-pandemic timepoint(s) (e.g., Xu et al.), our approach instead relied on predictive time series models that were trained on years of pre-pandemic data to make more robust counterfactual estimates of utilization expected in the absence of the pandemic:

L390-394: However, the utilization patterns described in our national study not only encompass a wider spectrum of granular conditions, **include** more pandemic subperiods **for analysis**, and **reflect comparisons from counterfactual utilization levels estimated via predictive models trained on** multiple years of pre-pandemic data, but also reveal relationships between disease attributes and utilization.

The last paragraph about the utility of this data. I suggest removing policy makers. This is a slippery slope in medicine, especially with what is being done to in obstetrics.

We understand this concern and appreciate the suggestion. We have removed "policymakers" from our discussion regarding the utility of the study's findings.

L466-469: An exploratory but expansive characterization of care utilization patterns across a specialty may serve as an initial tool for researchers and clinicians to evaluate the extent, magnitude, and differential impact of these disruptions.

Overall, I think this is an interesting concept which may help guide directions of care in the future. I think the database used is not the best representation of ophthalmic care given the limitations and practices included. Perhaps using on a more homogeneous population would give more accurate results for ophthalmology, but from a purely methodologic standpoint this paper does provide a foundation for future work.

We are very grateful for the reviewer's insightful feedback and critical assessment of the manuscript, which have strengthened this work considerably.

Reviewer #2 (Remarks to the Author):

This is a very interesting study, using nation-wide eye care registry (AAO IRIS@ Registry) data involving 44.62 million patients and 2,455 practices, the authors investigated how the restriction of COVID-19 pandemic may affect the utilisation of ophthalmic care. Despite the fact that COVID-19 restriction has been eased in majority of the countries in the world, the study findings are far-reaching because it provides evidence to demonstrate how restriction of infectious disease or lock down may cause substantial disruption of care in the context of ophthalmic care, a specialist care with high elasticity of demand. This massive dataset allows a meaningful investigation on eye care utilisation across a board range of diagnoses (3400 ophthalmic ICD-10 diagnosis) using longitudinal electronic health record data from a large national disease registry.

We thank the reviewer for their helpful suggestions and comments. We address each as described below, and the work has been made significantly stronger thanks to their critical remarks.

The primary outcome of the study was deviation from expected care utilisation for each diagnosis entity calculated as relative difference between observed and expected numbers of patients for a given month. It is understandable that an accurate estimation on the expected number of patients is the key for the estimation on deviation because the observed number of patients during pandemic should be easy to observe. In the manuscript, it is difficult to understand how these expected numbers of patient for a given month were calculated from the description in Page 8 / Line 173-179. Page 7 / Line 155 to 171 explained how the counterfactual models ("step 2") were used to generate a linear trend over years, but it is hard to understand how the expected numbers of patients were calculated before the pandemic?

We appreciate the need for clarification. For this study, we used models that were trained on years of pre-pandemic data to make counterfactual predictions of care utilization that would be expected, for each condition, in the absence of the pandemic. To do so, we first defined a common set of 10 candidate predictive generalized linear models (the full specifications of which are detailed in Text S1) that included various combinations of terms for seasonality and a linear trend over years ("Step 2A" in Figure 1). For each condition, we identified the best-fitting model among the ten candidate models through a leave-out-one-year blocked cross validation procedure ("Step 2B" in Figure 1), which

measures how well each of the ten candidate models is fit to pre-pandemic data (2017-2019) by selecting two pre-pandemic years at a time (e.g., 2017 and 2018) to serve as the training data used to fit the candidate model, and then comparing how closely the expected monthly numbers of patients predicted from the candidate model matches the actual numbers of observed patients in a holdout pre-pandemic year (e.g., 2019) that serves as the "test" (i.e., validation) set. The best-fit model would therefore be designated as the candidate model that makes the most accurate predictions across all holdout years (i.e., holdout year 2019 for training years 2017 and 2018; holdout year 2018 for training years 2017 and 2019; holdout year 2017 for training years 2018 and 2019), which was quantified by calculating the average of its predictive performance (measured via mean squared error) across all three holdout years. Finally, for each condition, the best-fit model was used to predict the counterfactual (i.e., "expected") numbers of patients during the pandemic study period (2020-2021) ("Step 2C" in Figure 1).

In the revision, we have edited the text to improve clarity and have also added clarifying remarks to the caption of Figure 1 (L695-697):

L190-194: In particular, for each condition, we used a leave-out-one-year blocked cross validation algorithm to select a model, among a prespecified set of candidate models, that had the best predictive ability, which was determined by identifying the candidate model with the lowest average mean squared error across all holdout years from the pre-pandemic period.

L198-202: Finally, to establish counterfactual levels of care utilization during the pandemic period, errors in the fit of the model selected for each diagnosis entity were assessed for overdispersion, after which 100,000 Monte Carlo-simulated predictions were generated from conditional Poisson, over-dispersed Poisson, or negative binomial distributions in accordance with the assessed mean-variance relationship of model predictions to residual values.

Also how does the coverage of IRIS registry (I assume the coverage was improving over time) may contribute to the bias on estimating the expected numbers of service?

The growth of the number of practices included in the IRIS Registry was mostly in its early years, with more stable growth in recent years (2020 and later). If a practice joined in the more recent years, data would be requested beginning from 2013 or the start date of their EHR system.

However (as also noted above in response to Reviewer #1), for this study (as noted in Lines 151-153 of the original manuscript), we only considered data from practices that contributed records to the IRIS Registry database for each month of the global study period (2017-2021). The findings from this analysis should therefore not reflect changes in overall utilization trends that may be attributable to practices that newly opened, became permanently closed, or experienced any other changes to their data reporting status to the IRIS Registry during the study period. Furthermore, our sensitivity analysis (Text S3) demonstrates no major changes in the primary outcome (estimated deviations from expected utilization levels), with or without requiring that the study be limited to

data from practices that were consistently active for each month from 2017-2021. To improve clarity, we have edited the text:

L181-184: Furthermore, to exclude from consideration changes in utilization trends that may be attributable to new openings, permanent closures, or other changes in the data reporting statuses of practices, we only analyzed EHRs from practices that reported data to the IRIS Registry throughout all months of the global study period.

The secondary outcome of the study was to examine the time used to recover to expected values and further classified into sustained recovery and partial recovery., but this part of results was not described in sufficient details in the manuscript.

Thank you for this very helpful point. After applying adjustments to the estimated p-values for each monthly deviation to correct for multiple testing (per Reviewer #3's suggestion in Point #11), we have accordingly re-computed time-to-recovery and recovery status for each diagnosis entity, and have now added expanded descriptions of these secondary outcomes in the Results section:

L364-381: Among all diagnosis entities, a broader set of conditions (116/261 = 44%) experienced some form of recovery (Table S6); however, many of these recoveries were not sustained (66/116 = 57%). The diagnosis categories with the highest proportions of conditions that experienced recoveries in utilization were uveitis and ocular inflammation (12/15 = 80%), post-operative complications (4/5 = 80%), ocular globe injuries/intraocular foreign bodies (3/4 = 75%), and cataract and other lens disorders (5/7 = 71%) (Figure 6). On the other hand, the diagnosis categories with the lowest proportions of conditions that experienced recovery were "other specified eye disorders" (2/13 = 15.4%), followed by refractive error (2/7 = 29%), retinal and vitreous conditions (28/75 = 37%), blindness and vision defects (5/13 = 38%), and cornea and external disease (22/56 = 39%) (Figure 6). Approximately half of all conditions in the diagnosis categories of oculofacial plastics and orbital conditions (15/29 = 51.7%), neuro-ophthalmology (8/16 = 50%), strabismus (3/6 = 50%), and glaucoma (7/15 = 47%) experienced recovery (Figure 6). Among all diagnosis entities that experienced partial or full recovery, the most common month at which recovery occurred was June 2020 (42/116 = 36.2%), followed by September 2020 (17/116 = 14.7%), December 2020 (14/116 = 12.1%), June 2021 (14/116 = 12.1%), and February 2021 (13/116 = 11.2%) (Figure 6).

We have also added more specific characterizations of conditions that decreased or increased the most in the post-hiatus phase (as reported in Tables S5A and S5B, respectively), and have summarized the recovery statuses of these conditions that experienced the most intense reductions, or rebounds, in utilization:

L327-349: Patterns of longitudinal deviations in care utilization across all diagnoses are summarized using a cluster heatmap of quarterly post-hiatus deviations (Figure 6), juxtaposed with April 2020 deviations, model performance errors, and time-to-recovery. We identified **33 conditions that experienced the most intense utilization reductions in the post-hiatus phase, defined as having an average monthly decrease in**

utilization of 20% or more over this period that was statistically significant (i.e., $\delta_{PH} \leq -0.20$ with $p \leq 0.05$; also represented by dark shades of red in the circular heatmap of Figure 6); many of these conditions were asymptomatic, slowly progressing, and/or NVT (Table S5A). The diagnosis categories most represented in this set of conditions with the largest post-hiatus utilization reductions were cornea and external diseases (e.g., conjunctivitis-related diagnoses, peripheral corneal degeneration), followed by retinal and vitreous conditions (e.g., retinal microaneurysms, unspecified background retinopathy, venous engorgement, and "other retinal microvascular abnormalities"), oculofacial plastics and orbital conditions (e.g., in situ carcinoma of the eye, benign eyelid neoplasm, orbital floor fracture, and "other eyelid degenerative disorders"), and blindness and vision defects (e.g., visual loss, suspect amblyopia, and color vision deficiencies) (Table S5A). Conjunctivitis-related diagnoses were particularly well-represented among the set of conditions that exhibited intense post-hiatus utilization reductions, with presentations for infectious keratoconjunctivitis decreasing the most ($\delta_{PH} = -0.38$, 95% CI: -0.41 to -0.35, $p < 0.001$) among all diagnosis entities. No conditions that had a mean post-hiatus utilization reduction of 20% or more also recovered partially or fully, except for the diagnosis of eyelid/periorcular superficial injury ($\delta_{PH} = -0.21$, 95% CI: -0.24 to -0.17, $p < 0.001$), which experienced a partial recovery in November 2020.

L351-362: Few conditions (15/261 = 6%) met or exceeded counterfactual utilization predictions in the post-hiatus period (i.e., $\delta_{PH} \geq 0$) (Table S5B); but among those that did, many were retinal and/or pediatric diseases, like unspecified DR with ($\delta_{PH} = 0.46$, 95% CI: 0.37 to 0.55, $p < 0.001$) and without ($\delta_{PH} = 0.04$, 95% CI: -0.01 to 0.09, $p = 0.11$) diabetic macular edema, infantile/juvenile cataract ($\delta_{PH} = 0.17$, 95% CI: 0.12 to 0.21, $p < 0.001$), eye injuries such as corrosion of the cornea/conjunctival sac ($\delta_{PH} = 0.14$, 95% CI: 0.08 to 0.21, $p < 0.001$) and ocular laceration without prolapse ($\delta_{PH} = 0.09$, 95% CI: 0.04 to 0.14, $p < 0.001$), and various stages of retinopathy of prematurity (ROP): ROP stage 3 ($\delta_{PH} = 0.12$, 95% CI: 0.06 to 0.18, $p < 0.001$), ROP stage 2 ($\delta_{PH} = 0.04$, 95% CI: -0.02 to 0.11, $p = 0.17$), and ROP with unspecified stage ($\delta_{PH} = 0.05$, 95% CI: 0.00 to 0.10, $p = 0.07$). All 15 diagnosis entities that met or exceeded post-hiatus counterfactual utilization levels also experienced recovery, with most of these conditions (12/15 = 80%) fully recovering.

Elasticities of disease diagnosis categories: the authors described the diseases as more elastic if greater negative deviations were observed during the pandemic and in fact observed greater decrease among the disease with less severe conditions. As expected, they observed the emergency eye diseases such as ocular globe injuries or intraocular foreign body as the least elastic diagnosis whereas the less severe eye diseases, such as refractive error, strabismus were more elastic.

We agree and thank the reviewer for this comment. However, as noted in Points #3-#4 from Reviewer #3, our usage of "elasticity" in this paper may be unconventional and possibly confusing, since some readers may be more accustomed to interpreting degrees of elasticity as formally defined in other contexts and subject areas (e.g., economics, engineering). To improve clarity, we have now edited our manuscript title and text to replace "elasticity"-related terms with more precise phrases.

Line 273-280 the authors described 36 ocular emergencies and their deviation during pandemic, hiatus or post-hiatus period, in the context of severity ranking. This is interesting also. Something missing in the analysis of deviation is about the eye diseases that require continuous or regular follow-up and care, for example, glaucoma or anti-VEGF AMD patients would require regular follow-up visit and treatment, it would be good to observe how elasticity may be associated with the diseases at different level of requirement on regular follow up and treatment.

We appreciate the reviewer's valuable suggestion. Our study was designed to comprehensively examine changes in care utilization for a wide range of ophthalmic diagnoses during the COVID-19 pandemic, providing insights into general trends and variations in the responsiveness of utilization patterns across diverse conditions. Given the extensive number of diagnosis entities analyzed, we felt that it was not feasible to delve into specific subsets of conditions in-depth, including those requiring continuous follow-up.

Similarly, while we agree that exploring the association between pandemic utilization trends and varying levels of follow-up would be informative, investigating such an association at a comprehensive scale may require defining robust metrics or indices to accurately quantify the degree of follow-up required for each diagnosis, across a sizeable number of conditions (e.g., similar to the set of 36 ocular emergencies we examined in Figure 5A by utilizing external measures of disease severity that were developed and validated by a separate study⁹). Quantifying the degree of follow-up required would require careful consideration and validation of metrics, necessitating a focused investigation beyond the scope of our current study.

In the revised discussion, we now highlight the degree of required follow-up as an additional dimension of analysis that can be further explored by future studies:

L399-405: We observed an inverse relationship between a condition's severity and its magnitude of underutilization during the pandemic, but there was heterogeneity in the strength and endurance of this relationship across different sets of diagnosis entities. This invites investigation into other factors to explain why some conditions may have been prioritized over others. Exploring associations between pandemic utilization levels and other attributes of diagnosis entities, such as the degree to which regular follow-up and clinical visits are required, could provide additional insights.

Reviewer #3 (Remarks to the Author):

Reviewer comments on Li et al 2022 Nature submission "Elasticities of Care Utilization Patterns During the COVID-19 Pandemic"

1. This paper is a clearly written and carefully documented discussion of the pattern of services for ophthalmic conditions in the US from 2017 through 2021. The study documents both the empirical magnitudes of reductions and the variation around expected trend values during the

immediate COVID-19 hiatus period (April 2020) and the post-hiatus recovery period. The authors call the percentage reduction in spending during each period the “elasticity of care” and show that some diagnoses were much more elastic (i.e., reduced much more) than others, and in other cases they were much less so.

We thank the reviewer for their close read, valuable comments, and critical remarks, which have strengthened our revised manuscript considerably.

2. I have two main concerns about the paper, one about the paper’s clarity and the other about whether the paper answers the all-important “so what” question.

We understand both concerns and address each as described below.

3. The authors use the term elasticities throughout the paper text without clearly defining what it is intended to measure. The concept of elasticities is used by economists, physicists and others to measure responsiveness with respect to some parameter or variable change, such as changes in income, price, or shock. I am unused to speaking of the elasticity of a variable’s response to an event, such as the pandemic, and found this usage in the paper confusing.

Thank you for raising this point, and we agree. We adopted the concept of elasticity from its traditional usage in disciplines like economics to frame our examination of possible factors driving differential care underutilization among the diagnoses we studied. The term “elasticity” seemed like an intuitive way to describe the varying degrees to which utilization for different health services were sensitive to the resource constraints or behavioral modifications imposed by the pandemic; however, we also realize that its usage in this context is unconventional and may cause confusion. We have therefore revised the text to replace “elasticity”-related terms with more precise descriptions.

L88-99: Previous studies by health economists have formally estimated the responsiveness, or elasticity, of demand of healthcare services to changes in cost or income. For instance, emergency room visits tended to exhibit little change in demand in response to changes in price, whereas pharmaceuticals, mental health/substance abuse treatment, and specialist care had high elasticities of demand. **Similarly, we explored how utilization levels for a wide range of ocular diagnoses exhibited varying degrees of sensitivity to possible pandemic-related restrictions to the seeking or delivery of care (e.g., resource constraints, behavioral modifications). We specifically investigated possible factors driving the differential underutilization of ophthalmic care during the pandemic by examining whether characteristics of medical problems themselves—namely, disease severity—were associated with observed changes in care utilization relative to levels expected in the absence of the pandemic.**

In the results and discussion sections, we have similarly substituted “elasticity”-related terms (including confusing expressions like “gradients of elasticity”) with more precise descriptions (L271-286, L288-315, L395-405).

4. It is notable that the abstract does not mention the word elasticity, even though it is in the paper title. I find the text clear at line 47 “Less severe conditions experienced greater utilization

reductions ..." and at line 49 they speak of "intense utilization reductions", rather than using the term elasticities. The paper title and text could be revised to be clearer on this.

We concur, and have revised the manuscript title accordingly:

L1-2: Shifts in Care Utilization Patterns During the COVID-19 Pandemic: A High-Dimensional Study of Presentations for Ophthalmic Conditions in the US

The text has also been similarly revised, as described in our response to the preceding Point #3.

5. My second concern is that it is unsurprising that there is heterogeneity in responsiveness of specific diagnoses to COVID-19 across the over 300 diagnosis entities examined. The authors admit this in their discussion section where they write

Lines 309-313: "While the presence of an elasticity gradient that corresponds to condition severity may be unsurprising (particularly during the hiatus), the heterogeneous strength and endurance of such a gradient across different sets of diagnoses invites future research to elucidate additional reasons to explain why some conditions may have been prioritized over others."

The authors also note:

326 "Further research is needed to understand the distinction between
327 benign reductions in visits from potentially harmful ones..."

Thank you for raising this concern. Our intention was not to assert novelty in the observation that there were variations in how different conditions responded to pandemic-related disruptions (or to present it as a central finding) but instead wanted to highlight the need for a fuller understanding of factors other than disease severity that could influence the prioritization of certain conditions. We have rephrased the text to better convey this point.

L399-405: We observed an inverse relationship between a condition's severity and its magnitude of underutilization during the pandemic, but there was heterogeneity in the strength and endurance of this relationship across different sets of diagnosis entities. This invites investigation into other factors to explain why some conditions may have been prioritized over others. For instance, exploring associations between pandemic utilization levels and other attributes of diagnosis entities, such as the degree to which regular follow-up and clinical visits are required, could provide additional insights.

As described in our response to Reviewer #2's last point, given the scale and scope of our analysis, we believe that adding additional dimensions of investigation (or focused cohort studies to establish the clinical impacts of missed or delayed care) would not be feasible, or optimal, to include in our current study, which is intended to serve as an exploratory but expansive characterization of pandemic care utilization patterns in ophthalmology.

6. While I agree that the paper has shown an association between both the severity and vision threatening/non-vision threatening diagnoses and their magnitudes of hiatus and post-hiatus reduction in frequency, the paper does not attempt to demonstrate the longer term significance of this variation. Are there more patients developing more serious conditions, (e.g., cancers, loss of vision, or blindness) because of the reduction in diagnosis? This is all left to future research. This left me unsatisfied as a reader.

We appreciate this point. While examining the clinical significance of missed or delayed care is undoubtedly important, it requires a more focused investigation that goes beyond the scope of our study. The robust assessment of long-term outcomes requires constructing separate patient cohort(s) that are customized to the condition(s) of focus, with tailored inclusion/exclusion criteria (e.g., excluding those with prior histories of certain conditions or co-morbidities, imposing minimum follow-up and look-back periods, requiring certain regimens and frequencies of treatment prior to being lost to follow-up), index event or exposure definitions (e.g., establishing what qualifies as being lost to follow-up),¹⁰ and outcome definitions (e.g., disease progression, presence of additional complications, visual acuity-based criteria). We therefore feel that this is not feasible to address in the current study, given that investigating long-term outcomes would require a different and separate study design, but have made minor edits to the text to better convey this perspective:

L430-437: Further research, such as longitudinal cohort studies designed to investigate the long-term clinical impact of missed or delayed care for specific condition(s) of focus, is needed to understand the distinction between benign reductions in visits from potentially harmful ones like decreases in screenings for common conditions that can lead to irreversible vision loss. Additionally, developing condition-specific definitions of loss to follow-up (LTFU) based on clinical considerations such as disease pathology and the appropriate frequency of follow-up or treatment,⁴⁰ enables targeted cohort studies to shed light on the patient-level determinants of LTFU during the COVID-19 pandemic.

7. The central contribution of the paper would seem to be the multicolor Figure 6 which illustrates for all of the 261 diagnostic clusters seven different dimension of information about the type of the impact of COVID-19. I found this figure overly complex, confusing and not particularly informative, as the authors implicitly reveal in the fact that they discuss this complex figure for only five lines, at lines 283-287. Perhaps extracts from the extensive supplement tables on which it is based would be more informative.

Thank you for this critique. We appreciate your suggestion about possibly replacing Figure 6 with information from the supplementary tables (e.g., Table S5). However, we think that Figure 6 remains an important part of our manuscript because it provides a multidimensional portrayal of pandemic care utilization patterns over time, a key objective of our study. We used a circular heatmap (often referred to as a "circos" plot)¹¹ to summarize utilization trends for all 261 diagnosis entities across 13 categories, and across multiple time periods. This approach allows for a compact, yet comprehensive, visualization of a vast quantity of information that would be challenging to convey in a linear format. Its advantages include the capacity to portray patterns of high-dimensional information across many categories through a single circular cluster heatmap (in contrast to the large linear heatmap shown in Figure S7, which spans several pages) and

the ability to convey different types of information (i.e., time-to-recovery, type of recovery, the magnitudes and directions of deviations at multiple time periods, and the magnitudes of counterfactual model performance errors) through multiple tracks (rings) focusing on the same object (i.e., a diagnosis entity, the unit of analysis for our study).

We believe this informational density, although visually complex, highlights interpretable high-level insights (which we now discuss in greater detail in the revision, as described in response to Point #8 below) while also preserving a level of granularity about the longitudinal utilization patterns of individual conditions that interested readers can learn more about through the supplemental information.

8. After reading the supplement, I found table S5 more interesting, with the high and low outliers for post-hiatus changes.

We thank the reviewer for this comment. To address this helpful point and Reviewer #2's similar suggestion above, in the revised text, we have now expanded our descriptions of diagnosis entities that experienced the most and least intense utilization reductions in the post-hiatus phase (L327-349, L351-362), and have also added further discussion of insights from Figure 6, including patterns of the time-to-recovery secondary outcome (L364-381).

9. I would have liked to have seen more discussion, perhaps in the supplement, about how the entities were merged from diagnoses. They are clearly much finer than the AHRQ CCSR. They represent a lot of work by someone. Perhaps in a new supplement text S5?

This is a very helpful point, and we now provide further discussion on the construction of diagnosis entities and diagnosis categories in a supplemental Text S4. We have also amended the main text to point the reader to Text S4 for further information.

L144-149: We followed a common set of considerations to modify these categorizations of ICD-10 codes where needed for the analytic purposes of this study, and to further create more granular groupings of these ICD-10 codes into diagnosis entities (Text S4). For instance, all diagnosis entities were required to have a sufficient utilization level (i.e., above single-digit patient counts) for each month of the study period (January 2017 to December 2021).

10. The paper should acknowledge online 124 the Agency for Healthcare Research and Quality Clinical Classification Software Revised (CCSR) version 2020.2.

We have now added this information to the description of the CCSR in the main text.

L141-144: We constructed an expansive inventory of ocular conditions to study by grouping more than 3,400 ophthalmic ICD-10 diagnosis codes into 336 clinically meaningful **diagnosis** entities adapted from **categorizations provided by the US Agency for Healthcare Research and Quality Clinical Classifications Software Refined (CCSR)** database version 2020.2.

11. The statistical methods used in the paper are of high quality and are clearly explained. The one issue I do not recall seeing discussed were corrections to t-statistics to account for the very large number of tests being conducted.

Thank you for this suggestion. We have now applied FDR corrections to adjust for multiple hypothesis tests conducted among the sets of estimated deviations (both on a monthly and quarterly timescale) for each diagnosis entity:

L245-248: To control for multiple testing among estimated monthly or quarterly deviations for each diagnosis entity, we calculated false discovery rate (FDR)-adjusted p-values for all monthly and quarterly deviations using the Benjamini-Hochberg method, with the FDR threshold set at 0.05.

These FDR-adjusted p-values have been used to update the results of the study's secondary outcome, time-to-recovery:

L222-225: As a secondary outcome, we examined the time it took for utilization to recover to expected values. Recovery was defined as three or more consecutive months for which **no statistically significant negative deviations from expectation ($\delta < 0$, adjusted $p \leq 0.05$) were recorded**. Among conditions that recovered, "sustained recovery" described those that did not experience further significant negative deviations, and "partial recovery" for the conditions that did.

This has resulted in a higher proportion of diagnosis entities experiencing partial or full recovery among all conditions studied (from $97/261 = 37\%$ previously to $116/261 = 44\%$ now). Furthermore, there are now more diagnosis entities that exhibit full, rather than partial, recovery; and similarly, some conditions now demonstrate an earlier recovery month than previously reported. These changes are reflected in the updated Table S5B (Pgs. 209-210 of the Supplement) and Table S6 (Pgs. 211-212 of the Supplement). In the revised main text, we have also provided an expanded description of the time-to-recovery outcome according to these latest results (L346-349, L360-362, L364-381).

We have also made small changes to the captions of Figures 2, 6, and S7, each of which contain heatmap visualizations, to reflect that the shading (darkness) of each heatmap cell is now a function of the product between the magnitude of the estimated deviation and the negative log of its *adjusted* p-value (instead of the negative log of its unadjusted p-value). There were no noticeable, or very modestly discernable, changes to these heatmap shadings after the incorporation of FDR-adjusted p-values.

Finally, we have added a new Table S8 in the supplement to report these FDR-adjusted p-values (Pgs. 227-238 of the Supplement) and have also amended the title of Table S7 to highlight that the p-values reported there are unadjusted.

12. At line 178 the paper notes bounds on $\delta = (\text{OBS}-\text{EXP})/\text{EXP}$ as

Where $\delta \in [-1, 1]$

I do not see why +1 is the upper bound of δ , since the OBServed value could be more than twice EXpected during a recovery period. We are not talking about elastic bands here.

Thank you for this catch – we agree and have removed the “ $\delta \in [-1, 1]$ ” condition (L212).

13. At line 229 the paper notes

229 On average, deviations were below expectation by 67%

230 (14%) in the nadir of the hiatus ($\delta_H = -0.67$), and by 13% (9%) post-hiatus ($\delta_{PH} = -0.13$)

(Figure

231 3B).

While this may be a true statement, it is not shown in Figure 3B. Perhaps add a mention in that table note?

We concur and have now added this information to the figure caption:

L724-727 (Caption for Figure 3B): The blue diamond represents the average of **all deviations for April 2020 (-0.67, standard deviation (SD): 0.14)** and the post-hiatus period **(-0.13, SD: 0.09)** across all 261 diagnosis entities.

14. Supplement table S1 reveals that ten candidate prediction models were examined for predicting baseline. The most flexible Model 10 uses 18 parameters to predict group means for only 36 time periods. The modeling also allowed three adjustments for overfitting. This almost surely is overfitting the data. No sensitivity analysis is presented about this degree of overfitting, only the chosen specification for each diagnosis entity. The sensitivity analysis presented in Text S3 does not mention this concern.

We appreciate the reviewer's remarks on this. While we acknowledge that complex models can be more susceptible to overfitting the training data, resulting in poor generalization abilities to unseen data, our usage of blocked cross-validation (specifically, the leave-out-one-year approach) for model selection was intended to guard against overfitting by penalizing candidate models that did not predict well on new data – i.e., contiguous, one-year blocks of pre-pandemic data that were used for validation. Blocked cross-validation has been shown to produce more reliable estimates of predictive error by accounting for temporal dependencies in the time series data;^{12,13} thus, so long as the processes underlying the counterfactual period are reasonably similar between the training and validation periods, the selected counterfactual model (which was identified as the one with the lowest cross-validated error) should not be overfit.

We further note that complex model specifications were infrequently selected as the best-fit counterfactual model among all 261 diagnosis entities in this study. For instance (as seen in Table S3), the most complex model specification (Model 8) was identified as the best-fit model for only six diagnosis entities (6/261 = 2%); in fact, counterfactual utilization levels for most diagnosis entities (202/261 = 77%) were predicted using the four most parsimonious model specifications (Models 1, 2, 3, and 4). Additional sensitivity analyses concerning model complexity may therefore be unlikely to yield meaningful discrepancies in overall study results.

In light of these considerations, we have not included further sensitivity analyses in the revised submission but would be glad to revisit this topic with additional editorial input.

15. It would make the tables more useful to viewers if the pdfs of the key table S2 included headers on each of its 100+ pages instead of only on the first page. This could use abbreviations for all variables so as to use only one line.) An Excel version could include the headers only at the top of the page.

Thank you for this suggestion. To improve readability, we have now included headers in the supplementary PDF on each page for Table S1, Table S2, and other tables that span multiple pages.

16. Table S1 contains the full crosswalk of 336 diagnostic entities considered in this paper, along with the binary flag of the 261 actually included in the analysis. The flags appear to be assigned to specific ICD-10-CM codes but are actually applied to the diagnosis entities.

The binary flag indicating whether the diagnosis entity was included for analysis has been reformatted to reflect its correspondence more clearly to diagnosis entities instead of specific ICD-10-CM codes.

Upon closer review, we also discovered that some rows (corresponding to unique ICD-10-CM codes) were truncated from the PDF version of Table S1. We regret this oversight, which has now been remedied in the revised Supplementary Information PDF.

17. This paper uses 13 categories of eye disorders, as listed in table S7 at the end, but these groups are not given a name there. Twelve of these are identical to the 12 CCSR categories included in the AHRQ CCSR version 2020.2 as linked on the AHRQ website included in the references to the main paper. The only difference appears to be that the authors have added one further category OGI/IOFB category which combines certain eye injury codes S05.XXX that the CCSR includes with other types of injuries. Also in that group are other foreign body in eye codes. Given the extreme reliance on the AHRQ system, you should acknowledge this reliance on CCSR better in the manuscript than the current passing reference on line 124.

We appreciate the need for further explanation. In the new Text S4, we now provide a fuller description of the 12 categories of eye-related diseases that we adopted from the CCSR and highlight the additional category of OGI/IOFB that we created for this study.

Pg. 5 of the Supplement: The full CCSR database contains groupings of more than 70,000 ICD-10-CM diagnosis codes into over 530 clinically meaningful categories; but for this study, we only considered ICD-10-CM codes that describe eye diseases and conditions by first examining the diagnosis codes attributed to the following 12 CCSR categories:

EYE001	Cornea and external disease
EYE002	Cataract and other lens disorders
EYE003	Glaucoma
EYE004	Uveitis and ocular inflammation

EYE005	Retinal and vitreous conditions
EYE006	Neuro-ophthalmology
EYE007	Strabismus
EYE008	Oculofacial plastics and orbital conditions
EYE009	Refractive error
EYE010	Blindness and vision defects
EYE011	Postprocedural or postoperative eye complication
EYE012	Other specified eye disorders

These CCSR categories are identical to 12 (out of 13 total) diagnosis categories used in this study; however, we created an additional diagnosis category of “ocular globe injuries/intraocular foreign bodies” (OGI/IOFB) to capture some eye injury diagnosis codes that were originally assigned to different (non-ophthalmic) categories of the CCSR than the ones listed above, as well as diagnosis codes that indicate the presence of a foreign body in the eye, which were previously assigned by the CCSR into the “Other specified eye disorders” (EYE012) category.

18. The mention of how the 336 diagnosis entities were created is inadequate. The text only includes the following:

124 All diagnosis

125 entities were required to have a sufficient utilization level (i.e., above single-digit patient counts)

126 for each month of the study period (January 2017 to December 2021).

This text should be expanded to indicate what information was made available to clinicians when grouping together individual diagnoses into your diagnostic entities.

Thank you for this suggestion – as noted in our response to Point #8, we have now edited the main text to reflect the addition of Text S4 (L144-147). In the revised text, we also further highlight that the “sufficient utilization” requirement is just one example of the criteria we considered when defining diagnosis entities.

L147-149: For instance, all diagnosis entities were required to have a sufficient utilization level (i.e., above single-digit patient counts) for each month of the study period (January 2017 to December 2021).

19. Also imprecise or misleading is this statement

126 Conditions with poor

127 counterfactual model performance (considered as $\geq 12.5\%$ root-mean squared percentage error

128 [RMSPE]) were also excluded, resulting in a final set of 261 diagnosis entities attributed to 13

129 mutually exclusive diagnosis categories (e.g., blindness and vision defects, cataract and

other

130 lens disorders, corneal and external conditions) (Table S1)

The number of entities excluded for the second reason (poor counterfactual model performance) appears to be those shown as having a RMSPE >12.5 as shown in table S2, not S1. A flag on these entities in that table would be informative.

We appreciate the need for clarity. In the revised supplement, we have added this binary flag to Table S2, along with a description of this indicator variable in the legends of Tables S1 and S2.

Furthermore, we have also edited the text to better convey that we excluded diagnosis entities for poor counterfactual model performance after defining the initial set of 336 entities according to a different set of considerations.

L149-151: After establishing an initial set of 336 diagnosis entities, to ensure that study findings were based on reliable predictions of utilization levels expected in the absence of the pandemic, conditions with poor counterfactual model performance (considered as $\geq 12.5\%$ root-mean-squared percentage error [RMSPE]) were also excluded, resulting in a final set of 261 diagnosis entities attributed to 13 mutually exclusive diagnosis categories (e.g., blindness and vision defects, cataract and other lens disorders, corneal and external disease) (Table S1) included for analysis in this study.

20. The index for the Supplement mentions but does not include the following on page 209.

References for the Supplementary Information Pg. 209

Thank you for pointing this out – the number of pages (208) in the version of the PDF stored on the journal's Manuscript Tracking System (MTS) did not match the length of the original document we had submitted (209 pages), likely due to automated reformatting by the MTS. We will work with the editorial office to ensure that the page numbers listed in the index correctly reflect the locations of all components of the revised Supplementary Information, which now has "References for the Supplementary Information" on Pg. 262.

Works Cited

1. Mehrotra, A. *et al.* The impact of COVID-19 on outpatient visits in 2020: Visits remained stable, despite a late surge in cases. *The Commonwealth Fund*
<https://www.commonwealthfund.org/publications/2021/feb/impact-covid-19-outpatient-visits-2020-visits-stable-despite-late-surge> (2021) doi:10.26099/BVHF-E411.
2. McKinsey & Co. *COVID-19 Hospital Insights Survey*.
<https://www.mckinsey.com/industries/healthcare-systems-and-services/our-insights/survey-us-hospital-patient-volumes-move-back-towards-2019-levels> (February, 18 2022).
3. McGough, M., Amin, K. & Cox, C. How has healthcare utilization changed since the pandemic? *Peterson-KFF Health System Tracker*
<https://www.healthsystemtracker.org/chart-collection/how-has-healthcare-utilization-changed-since-the-pandemic/> (2023).
4. American Academy of Ophthalmology. EHR Systems - IRIS® Registry. *American Academy of Ophthalmology* <https://www.aao.org/iris-registry/ehr-systems> (2023).
5. Gray, J. & Holmes, A. *Pandemic Aftershocks: Examining the Decline in Healthcare Utilization in California During COVID-19*. https://www.manifestmedex.org/wp-content/uploads/MX_Pandemic_Aftershocks_Examining_CA_Healthcare_Utilization_During_Covid19.pdf (2021).
6. Gillis, K. *Impacts of the COVID-19 Pandemic on 2020 Medicare Physician Spending*.
<https://www.ama-assn.org/system/files/2020-prp-covid-impact-medicare-physician-spending.pdf> (2021).
7. Williams, R. D., II, Shah, A., Doty, M. M., Fields, K. & FitzGerald, M. The Impact of COVID-19 on Older Adults. *The Commonwealth Fund*
<https://www.commonwealthfund.org/publications/surveys/2021/sep/impact-covid-19-older-adults> (2021) doi:10.26099/mqsp-1695.

8. Schuster, N. A. *et al.* Older adults report cancellation or avoidance of medical care during the COVID-19 pandemic: results from the Longitudinal Aging Study Amsterdam. *Eur. Geriatr. Med.* **12**, 1075–1083 (2021).
9. Bourges, J.-L., Boutron, I., Monnet, D. & Brézin, A. P. Consensus on Severity for Ocular Emergency: The BASic SEverity Score for Common Ocular Emergencies [BaSe SCOrE]. *J. Ophthalmol.* **2015**, 576983 (2015).
10. Khurana, R. N., Li, C. & Lum, F. Loss to Follow-up in Patients with Neovascular Age-Related Macular Degeneration Treated with Anti-VEGF Therapy in the United States in the IRIS® Registry. *Ophthalmology* **130**, 672–683 (2023).
11. Gu, Z., Gu, L., Eils, R., Schlesner, M. & Brors, B. circlize Implements and enhances circular visualization in R. *Bioinformatics* **30**, 2811–2812 (2014).
12. Bergmeir, C. & Benítez, J. M. On the use of cross-validation for time series predictor evaluation. *Inf. Sci.* **191**, 192–213 (2012).
13. Roberts, D. R. *et al.* Cross-validation strategies for data with temporal, spatial, hierarchical, or phylogenetic structure. *Ecography* **40**, 913–929 (2017).

Reviewers' comments:

Reviewer #1 (Remarks to the Author):

Sufficient changes have been made.

Reviewer #2 (Remarks to the Author):

The manuscript offers a lucid and well-substantiated analysis of the trends in services for ophthalmic conditions in the US from 2017 to 2021. The subject possesses considerable clinical relevance, furnishing a preliminary yet comprehensive depiction of care utilization patterns across this specialty. Such a characterization aids both researchers and clinicians in ascertaining the scope, intensity, and disparate effects of these disturbances. Below are some suggestions for refinement:

While the paper acknowledges the primary causes of visual impairment—namely, age-related macular degeneration, diabetic retinopathy, cataract, and glaucoma—it only delineates three, excluding cataract, in Table S4 and Figure 5B to represent varying VT/NVT statuses. It is imperative to elucidate the situation pertaining to cataract and its associated ophthalmic care utilization.

The narrative underscores the data's high-dimensional and high-resolution nature. Could the authors provide a succinct elucidation of this characterization?

It would be beneficial to delineate the strengths of this investigation in a distinct section, accentuating its unique contributions and advantages.

I suggest to discuss more about Figure 5B. This illustration conveys pivotal insights regarding routine check-ups for certain chronic eye ailments. Integrating data like the incidence or progression rates of the specific conditions might endow the study with enriched clinical implications, especially concerning follow-up protocols and distinguishing between benign and precarious lapses in follow-up.

Reviewer #3 (Remarks to the Author):

One confusing part for me is the brief use of the term “Longitudinal Deviation” as shown below. The phrase is used only twice, as shown here. One could think that the longitude you are studying is across diagnoses.

290 Identification of Clusters of Diagnosis Entities with Similar
Longitudinal Deviation Patterns

291 Patterns of longitudinal deviations in care utilization across all diagnoses are
292 summarized using a cluster heatmap of quarterly post-hiatus deviations (Figure 6), juxtaposed
293 with April 2020 deviations, model performance errors, and time-to-recovery.

This specific phrase is not used in Figure 6 which is being described. Perhaps you could change it to

“longitudinal decline and recovery patterns” or “inter-temporal deviation patterns” or even just “deviations over time”.

In a related way, it would be more specific if instead of “longitudinal utilization trends” to write “changes in utilization over time”. A variety of wordings for longitudinal or deviations are used in the paper for this concept. Standardization would be useful.

I now like Figure 6, which is better motivated and explained, although still complex. As a figure, one flaw is the lack of color gradation between the different levels of the initial decline. The clusters plotted in Figure 3 suggest that the shades of purple range only from about -0.3 to -0.9, so almost half of the allowed gradation (0 to -1) is not actually used. Stated differently, it is hard to pick out the diagnostic entities that are at the extremes. This might be fixed when JAMA Network redoes the figure and perhaps uses a darker color of purple for the upper extreme, or the range is trimmed to the sample min and max of the diagnostic entities rather than -1 to 0. I am looking at a 36-inch monitor when evaluating this figure, so it is not because I am using black and white paper copy or a small image. The figure is even less interpretable when printed out in black and white on normal sized paper.

The editors should decide whether notes on figures such as Figure 6 to the acronyms and abbreviations used are needed only once or on each figure. They are missing and not always consistent across figures and text.

I would insert the word DECLINES (or CHANGES) as shown here.

378 We also identified clusters of diagnoses that shared similar longitudinal utilization trends,
379 thereby highlighting conditions that had above- and below-average utilization DECLINES and
distinguishing
380 the relatively few conditions that experienced recovery from those that did not.

I could not figure out what the black dots are in Figure 5 and did not see them explained in the figure notes.

Reviewer Comments

Reviewer #1 (Remarks to the Author):

Sufficient changes have been made.

We thank the reviewer for their insightful feedback, which has been instrumental in enhancing the quality of our work.

Reviewer #2 (Remarks to the Author):

The manuscript offers a lucid and well-substantiated analysis of the trends in services for ophthalmic conditions in the US from 2017 to 2021. The subject possesses considerable clinical relevance, furnishing a preliminary yet comprehensive depiction of care utilization patterns across this specialty. Such a characterization aids both researchers and clinicians in ascertaining the scope, intensity, and disparate effects of these disturbances. Below are some suggestions for refinement:

We are grateful for the reviewer's close re-examination of the manuscript and important suggestions for refinement. We address each as described below.

While the paper acknowledges the primary causes of visual impairment—namely, age-related macular degeneration, diabetic retinopathy, cataract, and glaucoma—it only delineates three, excluding cataract, in Table S4 and Figure 5B to represent varying VT/NVT statuses. It is imperative to elucidate the situation pertaining to cataract and its associated ophthalmic care utilization.

Thank you for this comment. We did not include any cataract-related diagnosis entities in Figure 5B due to the difficulty of assigning VT and NVT labels to the different types of cataract diagnoses based on ICD-10 codes alone.

Age-related cataracts, the most common form of cataract, progress slowly (it often takes years before vision is affected),¹ and although some forms of age-related cataracts develop more quickly than others (e.g., posterior subcapsular cataracts have been shown to progress faster than nuclear and cortical cataracts)², referrals for cataract evaluation are broadly considered to be nonurgent,¹ as the presence of a cataract alone does not require urgent action unless there is a secondary complication resulting from the cataract (which is currently out of the study's scope to examine, as doing so would require the customized construction of patient cohorts).³

Some pediatric (infantile/juvenile) cataracts can be comparatively more urgent to address, depending in part on parameters of the cataract that are not consistently or comprehensively captured by ICD-10 codes (namely, its size and location), and not all pediatric cataract cases are considered urgent. Cataract extraction is needed within days to weeks of discovery to prevent irreversible amblyopia if the visual axis is obstructed and the eye's visual development, or overall health, is put at risk; but in other cases, follow-up visits for observation at varying frequencies (depending on patient characteristics, e.g., age) may be the more appropriate management approach.³ For instance, older children with cataracts are followed less frequently than infants and

younger children, because amblyopia is less likely to develop in spite of possible cataract progression.³

Thus, we are unable to assign VT vs. NVT labels feasibly or reliably to the different types of cataract diagnoses, unlike the subtypes of diabetic retinopathy, age-related macular degeneration, or glaucoma, which are more easily categorized as VT or NVT.^{3,4} In light of these considerations, we have not regenerated Figure 5B to include cataract diagnoses but appreciate the issue and need for more clarity around this.

The narrative underscores the data's high-dimensional and high-resolution nature. Could the authors provide a succinct elucidation of this characterization?

We agree with the need for further explanation and have edited the text accordingly:

L127-136: The high-resolution insights (i.e., derived from detailed ophthalmic diagnoses) generated from this high-dimensional analysis (i.e., conducted across an expansive range of ocular conditions, and over different subperiods and all months spanning the first two years of the pandemic) may inform future studies aiming to determine the clinical impacts of missed or delayed care in specific patient populations and disease cohorts, monitor the pandemic's effect on healthcare access, and clarify distinctions between harmful and benign reductions to care.

It would be beneficial to delineate the strengths of this investigation in a distinct section, accentuating its unique contributions and advantages.

We concur, and have re-written the first paragraph of the discussion to review the contributions of this study more clearly and with greater detail:

L393-414 (superscripts denote reference numbers in the main text): Our study presents a comprehensive exploratory analysis of care utilization patterns for ophthalmic diagnoses during the first two years of the COVID-19 pandemic. Although the prioritization of care for more severe conditions in the early pandemic phase has been previously reported across other specialties³³ and among select ophthalmology practices,³⁴⁻³⁶ this study contains distinct advantages over previous research. First, unlike prior investigations that only focused on broad diagnostic categories or a limited range of clinical problems, the wide spectrum of 261 granular ophthalmic conditions included in this analysis provides both an expansive and detailed view into the evolving visit patterns for diagnoses spanning a single specialty. Second, our analysis included multiple subperiods of the pandemic extending beyond its acute phase. This broad temporal scope facilitated versatile explorations of utilization patterns over time, including comparisons of how deviations from expected utilization levels varied over different subperiods for the same diagnosis entity or category, and measurements of the time it took for conditions to reach or exceed these counterfactual expectations. Another key strength of our study lies in the usage of predictive models trained on multiple years of pre-pandemic data to establish counterfactual utilization levels for each diagnosis entity, which provides a robust baseline for inferring the effect of the pandemic on care utilization. Furthermore, this investigation also examines how changes in care utilization during the pandemic

may be related to attributes of diseases themselves, providing insights into characteristics associated with conditions that may have been prioritized during the pandemic. Collectively, the contributions of our analysis address several key gaps in the existing literature of pandemic utilization studies³³ and help advance a more thorough understanding of pandemic-driven shifts in care utilization for ocular conditions.

I suggest to discuss more about Figure 5B. This illustration conveys pivotal insights regarding routine check-ups for certain chronic eye ailments. Integrating data like the incidence or progression rates of the specific conditions might endow the study with enriched clinical implications, especially concerning follow-up protocols and distinguishing between benign and precarious lapses in follow-up.

Thank you for this suggestion. In the revision, we now provide an expanded discussion of the potential public health implications of the sustained decreases in utilization we observed for diagnoses corresponding to the leading causes of visual morbidity that are highlighted in Figure 5B and elsewhere (e.g., Figure 2 and Figure 4A).

L447-454 (superscripts denote reference numbers in the main text): Throughout the pandemic study period, we noted a consistent decrease in visits related to the leading causes of blindness and vision loss among adults in the United States²⁸ and globally:³⁹ age-related macular degeneration, cataract, diabetic retinopathy, and glaucoma. The early detection and timely treatment of these conditions are largely cost-effective and efficacious at restoring, or preventing further deterioration of, vision.³⁹ The burdens of these conditions are projected to increase due to ageing of the population (AMD, cataract, glaucoma) and global increases in diabetes prevalence (DR),³⁹ underscoring the importance of considering the potential population health repercussions of these observed visit reductions.

We subsequently describe the population burden estimates (both globally and in the U.S.), general management approaches, and the potential clinical consequences of missed screening and/or follow-up visits for age-related macular degeneration (L456-466), cataract (L468-482), diabetic retinopathy (L484-503), and glaucoma (L505-520).

Reviewer #3 (Remarks to the Author):

One confusing part for me is the brief use of the term "Longitudinal Deviation" as shown below. The phrase is used only twice, as shown here. One could think that the longitude you are studying is across diagnoses.

290 Identification of Clusters of Diagnosis Entities with Similar Longitudinal Deviation Patterns

291 Patterns of longitudinal deviations in care utilization across all diagnoses are
292 summarized using a cluster heatmap of quarterly post-hiatus deviations (Figure 6),
juxtaposed

293 with April 2020 deviations, model performance errors, and time-to-recovery.

This specific phrase is not used in Figure 6 which is being described. Perhaps you could change

it to "longitudinal decline and recovery patterns" or "inter-temporal deviation patterns" or even just "deviations over time".

Thank you for raising this point – to avoid the potential confusion, we have removed the term "longitudinal" (L118, L575, L906) or replaced it with "over time" for better clarity (L233-234, L330-331, L332, L523-525).

In a related way, it would be more specific if instead of "longitudinal utilization trends" to write "changes in utilization over time". A variety of wordings for longitudinal or deviations are used in the paper for this concept. Standardization would be useful.

We appreciate the need for standardization and have made the needed changes as described above.

I now like Figure 6, which is better motivated and explained, although still complex. As a figure, one flaw is the lack of color gradation between the different levels of the initial decline. The clusters plotted in Figure 3 suggest that the shades of purple range only from about -0.3 to -0.9, so almost half of the allowed gradation (0 to -1) is not actually used. Stated differently, it is hard to pick out the diagnostic entities that are at the extremes. This might be fixed when JAMA Network redoes the figure and perhaps uses a darker color of purple for the upper extreme, or the range is trimmed to the sample min and max of the diagnostic entities rather than -1 to 0. I am looking at a 36-inch monitor when evaluating this figure, so it is not because I am using black and white paper copy or a small image. The figure is even less interpretable when printed out in black and white on normal sized paper.

We understand this concern. To provide maximal contrast, we have accordingly restricted the range of the color gradient for the purple ring to be bounded by the maximum decrease in utilization among all diagnosis entities during April 2020 ($\delta_{\max} = -0.91$), which is also now represented using a darker shade of purple, and the minimum decrease in utilization ($\delta_{\min} = -0.18$), which is now represented using a nearly white color.

The editors should decide whether notes on figures such as Figure 6 to the acronyms and abbreviations used are needed only once or on each figure. They are missing and not always consistent across figures and text.

We regret this oversight and have closely revisited and updated all figure captions (including that of Figure 6) to comprehensively and consistently delineate all abbreviations used; additions or changes are now reflected at L833-835 (for Figure 1), L849-851 (for Figure 2), L867-869 (for Figure 3), L880-883 (for Figure 4), L898-900 (for Figure 5), and L912-913 and L918-926 (for Figure 6). In the caption for Figure 6 and elsewhere, we focused on ensuring standardization of diagnosis categories names to exactly match the full category names listed in Text S4.

I would insert the word DECLINES (or CHANGES) as shown here.

378 We also identified clusters of diagnoses that shared similar longitudinal utilization trends,
379 thereby highlighting conditions that had above- and below-average utilization DECLINES
and distinguishing

380 the relatively few conditions that experienced recovery from those that did not.

Thank you for this catch – we have made the needed changes to this sentence, along with additional minor edits for improved clarity:

L523-527: we also identified clusters of diagnoses that **experienced exceptional deviations in utilization patterns over time**, thereby highlighting conditions that had above- and below-average utilization **changes** and distinguishing the relatively few conditions that experienced recovery from those that did not.

I could not figure out what the black dots are in Figure 5 and did not see them explained in the figure notes.

The black dots in the boxplots of Figure 5 represent individually-plotted outliers, defined as data points that are located at a distance greater than 1.5 times the Interquartile Range (IQR) from either the lower quartile (Q1) or the upper quartile (Q3) in a boxplot (https://ggplot2.tidyverse.org/reference/geom_boxplot.html).

The caption for Figure 5 has now been edited to include this explanation:

L887-900: Boxplots depicting distributions of deviations from expectation, during the hiatus (orange boxplots) and post-hiatus (blue boxplots) periods, stratified by (A) increasing levels of severity (derived from the aggregate BaSe SCOrE's compiled by Bourges et al.) for a set of 36 common ocular emergencies, and (B) vision-threatening (VT) vs. non-vision-threatening (NVT) status for age-related macular degeneration (AMD), diabetic retinopathy (DR), and glaucoma diagnoses (**Table S4**). **Outliers, indicated as black dots, are data points that are located at a distance greater than 1.5 times the interquartile range from either the lower quartile or the upper quartile of the boxplot. The 'whiskers' of the boxplots, which extend from the boxes as black vertical lines, represent the range of values that lie within 1.5 times the interquartile range from the lower and upper quartiles.** To test for statistically significant differences in the central tendencies between distributions of deviations, we used Kruskal-Wallis (A) and Mann-Whitney U (B) tests to compute p-values (gray text). Abbreviations: "BaSe SCOrE" = BAsic SEverity Score for Common OculaR Emergencies; "DR" = diabetic retinopathy; "AMD" = age-related macular degeneration; VT = vision-threatening; NVT = non-vision-threatening.

Works Cited

1. *Basic Ophthalmology: Essentials for Medical Students*. (American Academy of Ophthalmology, 1731).
2. Miller, K. M. *et al.* Cataract in the Adult Eye Preferred Practice Pattern. *Ophthalmology* **129**, P1–P126 (2022).
3. *The Wills Eye Manual: Office and Emergency Room Diagnosis and Treatment of Eye Disease*. (LWW, 2021).
4. Burton, M. J. *et al.* The Lancet Global Health Commission on Global Eye Health: vision beyond 2020. *Lancet Glob Health* **9**, e489–e551 (2021).

REVIEWERS' COMMENTS:

Reviewer #4 (Remarks to the Author):

Dear Authors,

I commend you on the thorough revisions made to your manuscript. The concerns previously raised by Reviewer #2 appear to have been thoughtfully and effectively addressed.

The COVID-19 pandemic has undeniably been a global crisis, exerting unprecedented pressure on health care systems worldwide. Notably, the strategies employed to manage this pandemic, ranging from public health measures to health care system adaptations and the prioritization of resources, have varied widely among different nations and communities. Your study, which utilizes a registry from the United States, offers valuable insights. However, it is crucial to clarify the scope of your findings, emphasizing that they may not be universally applicable to all countries. Your data represents one registry in one country.

In this context, I suggest making explicit references to the geographical limitation of your study, particularly in your abstract and conclusion. For instance, consider rephrasing to highlight the specific context of your research, as follows (with my suggested addition in capital letters):
"We derived high-resolution insights on pandemic care utilization IN THE UNITED STATES from high-dimensional data using an analytical framework that can be applied to study healthcare disruptions in other settings and inform efforts to pinpoint unmet clinical needs."

This clarification will not only enhance the accuracy of your study but also help readers understand the context and potential limitations of your findings.

Thank you for your dedication to advancing our understanding of healthcare utilization during the pandemic.

Sincerely

Reviewer Comments

Reviewer #4 (Remarks to the Author):

Dear Authors,

I commend you on the thorough revisions made to your manuscript. The concerns previously raised by Reviewer #2 appear to have been thoughtfully and effectively addressed.

The COVID-19 pandemic has undeniably been a global crisis, exerting unprecedented pressure on health care systems worldwide. Notably, the strategies employed to manage this pandemic, ranging from public health measures to health care system adaptations and the prioritization of resources, have varied widely among different nations and communities. Your study, which utilizes a registry from the United States, offers valuable insights. However, it is crucial to clarify the scope of your findings, emphasizing that they may not be universally applicable to all countries. Your data represents one registry in one country.

In this context, I suggest making explicit references to the geographical limitation of your study, particularly in your abstract and conclusion. For instance, consider rephrasing to highlight the specific context of your research, as follows (with my suggested addition in capital letters): "We derived high-resolution insights on pandemic care utilization **IN THE UNITED STATES** from high-dimensional data using an analytical framework that can be applied to study healthcare disruptions in other settings and inform efforts to pinpoint unmet clinical needs."

This clarification will not only enhance the accuracy of your study but also help readers understand the context and potential limitations of your findings.

Thank you for your dedication to advancing our understanding of healthcare utilization during the pandemic.

Sincerely

We thank the reviewer for their close read of our paper and the helpful comments. We appreciate the need to better clarify the geographical scope of our findings, and have edited the text accordingly in the Conclusions section of the abstract (L343-L345) and elsewhere (L613-614, L868-870).